



# Technical note: Theoretical and experimental investigation of isotopic exchange between water vapour and droplets under isothermal saturation conditions

Wenwen Bai[1,2,3,4], Jiahua Wei[1,2, 3,4,5], Sanchuan Ni[1,2,3,4], Zhanyu Yao[6], Yifeng Liu[1,2,3,4], Jingjing Ding[1,2,3,4], Kaiyu Wang[1,2,3,4], Nan Liu[1,2,3,4], Liner Wang[1,2,3,4], Miao Wu[1,2,3,4]

[1]School of Civil Engineering and Water Resources, Qinghai University, Xining, 810016, China

[2]Laboratory of Ecological Protection and High-Quality Development in the Upper Yellow River, Qinghai University, Xining, 810016, China

[3]Key Laboratory of Water Ecology Remediation and Protection at Headwater Regions of Big Rivers, Ministry of Water Resources, Qinghai University, Xining, 810016, China

[4]State Key Laboratory of Plateau Ecology and Agriculture, Qinghai University, Xining, 810016, China

[5]State Key Laboratory of Hydroscience and Hydraulic Engineering/Department of Hydraulic Engineering, Tsinghua University, Beijing, 100084, China

[6]Weather Modification Centre, China Meteorological Administration, Beijing, 100081, China

*Correspondence to*: Jiahua Wei (weijiahua@tsinghua.edu.cn)

**Abstract:** Water cycle process will deeply affect water vapour isotope composition, in addition to the evaporation and condensation processes, the exchange process is also a crucial process that influences isotopic variations. To explore the mechanism of isotope exchange under isothermal saturation conditions, we developed an isotope exchange calculation equation for water vapour (IECEWV) based on the conservation of exchange quantity and conducted indoor trials with 100 control groups to validate the IECEWV. The experimental findings demonstrated that the isotopic values of water vapour exhibited enrichment, depletion, and stabilisation during the exchange process when the droplet size first increased and then stabilised, because of the interaction between the exchange state, isotopic gradients, and specific surface area of the droplets according to IECEWV. The variation trend of IECEWV calculated isotopes in water vapour was consistent with experimental value, with average maximum relative errors of 1.66 % for $\delta^2$H and 3.19 % for $\delta^{18}$O, and IECEWV analysis further indicates isotopes exchange can lead to the hydrogen and oxygen isotope line of water vapour deviate from the origin. Furthermore, the linear relationship between hydrogen and oxygen isotopes in water vapour after exchanged can be characterised by linear clusters with the same slope, the *d*-excess of water vapour remains essentially constant. Future studies may combine the IECEWV with the Rayleigh fractionation model to explore the coordinated changes in precipitation and atmospheric water vapour isotopes.

## 1 Introduction

Water vapour is the most active hydrological component in the global water cycle (Dar et al., 2020; Galewsky et al., 2016),



constituting the direct source of precipitation (He et al., 2019; Qiang et al., 2019; Wang et al., 2016a), revealing that the isotopic characteristics and mechanistic variations of water vapour and precipitation during the precipitation process are important for comprehending the mechanisms of precipitation formation. The water vapour source is an important factor affecting isotopes, for instance, water vapour over the Qinghai-Tibet Plateau primarily originates from the Bay of Bengal and westerlies (Wang et al., 2021; Yao et al., 2013; Yu et al., 2016), manifesting characteristics of isotopic depletion and enrichment (Guo et al., 2017; Zhang et al., 2019), respectively. Through advection and entrainment, water vapour feeding the cloud can significantly affect the cloud droplet isotope composition (Spiegel et al., 2012a), in precipitation events governed by multiple water vapour sources (Xu et al., 2022), rapid shifts in the source of water vapour distinctly impact precipitation isotopic values (Xu et al., 2022) and vapour isotopic values owing to isotope fractionation. In addition, environmental factor variation will influence the isotopic process (Moreno et al., 2021; Santos et al., 2022) leading to isotope composition changing of water vapour, and researchers have further discerned "precipitation amount effects" and "temperature effects" (Dansgaard, 1964) in water vapour isotopes (Xue et al., 2023).

Water cycle process significantly affect water vapour isotopic values, in addition to condensation and evaporation, the exchange process also influences the isotopes of water vapour and the resulting precipitation. When precipitation occurs and the cloud layer temperature continues to decrease (Yang et al., 2011), condensation becomes the primary driver of isotopic changes (Gu et al., 2011). However, when the relative humidity reaches 100 %, if the isotopic gradient between the liquid water and vapour is not balanced, in addition to the condensation process, isotopic exchange (Bai et al., 2021b; Galewsky et al., 2016; Stewart, 1975) will occur between raindrops and the surrounding water vapour in the precipitation cloud layer. Studies have confirmed that exchange leads to further isotopic depletion of water vapour (Bai et al., 2022b; Bai et al., 2021a) and that the exchange process is related to the isotopic gradient, particle size, and exchange time (Friedman et al., 1962; Stewart, 1975), and when the isotopic balance is achieved, the isotopic value of the droplet is independent of its size (Spiegel et al., 2012b). With reference to the calculation equation, the changing process of heavy molecules can be determined, which subsequently yields the isotopic values of raindrops after exchange.

The linear relationship between hydrogen and oxygen isotopes has been widely studied in isotope hydrology, and exchange process may change the linear relationship between hydrogen and oxygen isotopes in precipitation and water vapour. Researchers have found that the slope of this linear relationship is predominantly controlled by equilibrium fractionation processes, whereas the intercept is mainly influenced by non-equilibrium fractionation processes (Gu et al., 2011). Considering the actual precipitation conditions, researchers have incorporated both equilibrium and non-equilibrium fractionation processes and derived theoretical relationships for the precipitation of hydrogen and oxygen isotopes (Wang et al., 2009; Wang et al., 2016b), the resulting slope and intercept aligned well with the global precipitation isotopic line (Craig, 1961). However, a





linear relationship between the hydrogen and oxygen isotopes of water vapour in an isotope exchange scenario has not yet been reported. Given the exchange between a large number of droplets and water vapour, it is urgent to use a new calculation equation to calculate and analyse the hydrogen and oxygen isotope values of water vapour and their linear relationships.

The difference in molecular activity due to the isotopic atomic mass is the main reason for the variation in isotopes between water vapour and precipitation. Owing to their disparate molecular activities, heavier molecules tend to exhibit relatively enriched concentrations in the liquid phase and depleted concentrations in the gaseous phase (Gu et al., 2011). This phenomenon is commonly referred to as isotopic fractionation. Under equilibrium fractionation conditions, the interphase isotopic differences can be described by the principle of mass conservation, where the fractionation factor remains constant

over time. However, non-equilibrium fractionation occurs predominantly in actual processes and primarily involves kinetic and diffusion fractionation (Gu et al., 2011). In many cases, the Rayleigh fractionation model is utilised to describe such nonequilibrium processes (Liu et al., 2015), providing a functional relationship between the isotopic ratios of condensed water vapour and residual water vapour. However, because of its inability to account for isotope exchange processes, it is necessary to derive an exchange calculation equation for water vapour isotopes based on mass-conservation principles to elucidate the

mechanisms underlying isotopic evolution during the exchange process.

In the actual scenario where evaporation and exchange occur simultaneously, the isotope exchange process leads to weaker equilibration on the water vapor and precipitation isotope values (Graf et al., 2019), but this is insufficient for quantitatively the impact of exchange on the water vapor isotope alone, due to the gap in the field particle size and relative humidity and the complexity of the process (Sarkar et al., 2023). Therefore, it is necessary to use theoretical and

experimental research methods to deeply understand the isotope exchange process. In this study, we established an isotopic exchange calculation equation for water vapour (IECEWV) under isothermal saturation conditions based on the principle of conservation of exchange quantity. To validate the IECEWV, we designed an experimental platform. Our experiment incorporated noncontact technology to manipulate droplet sizes (Bai et al., 2022a; Bai et al., 2020) while monitoring the isotopic characteristics of the original vapour and liquid water through sample collection. Subsequently, the observed isotopic

values were compared and analysed using theoretical calculations. The study will deeply our understanding of isotopic changes in the water cycle, especially for coordinated changes in precipitation and water vapour isotopes, and further expand the applicable accuracy and scope of the Rayleigh fractionation model.

The remainder of the paper is organised as follows: Section 2 derives the IECEWV under isothermal saturation conditions based on the principle of conservation of exchange quantity; analyses the theoretical equation, including the influence of each

factor on the theoretical calculation value and their linear relationship; and presents the experimental platform, covering the experimental apparatus design, experimental process, data acquisition, and data processing. Section 3 analyses the





experimental results and factors influencing the isotopic changes in water vapour, providing insights into droplet microphysics and the temporal variation of water vapour isotopes. Section 4 discusses the results and explores the calculations and comparative analyses of the temporal variations in the water vapour isotopic values during the experimental process,

establishes a linear relationship between $\delta^2$H and $\delta^{18}$O of water vapour based on the IECEWV; and summarises the limitations of the experimental conditions and IECEWV. Finally, Section 5 concludes the study.

## 2 Methods and materials

To explore the isotope variation of water vapour during the exchange process between droplets and water vapour, we derived the IECEWV based on the conversion of droplets and water vapour, which means that the exchange molecule quantity is equal,

but the exchanged molecule type is different, leading to a change in the isotope value of water vapour. The experiment was used to validate the IECEWV and the calculated isotopes were compared with the experimental monitoring values. The correlation between the theory and experiment is illustrated in Figure 1.

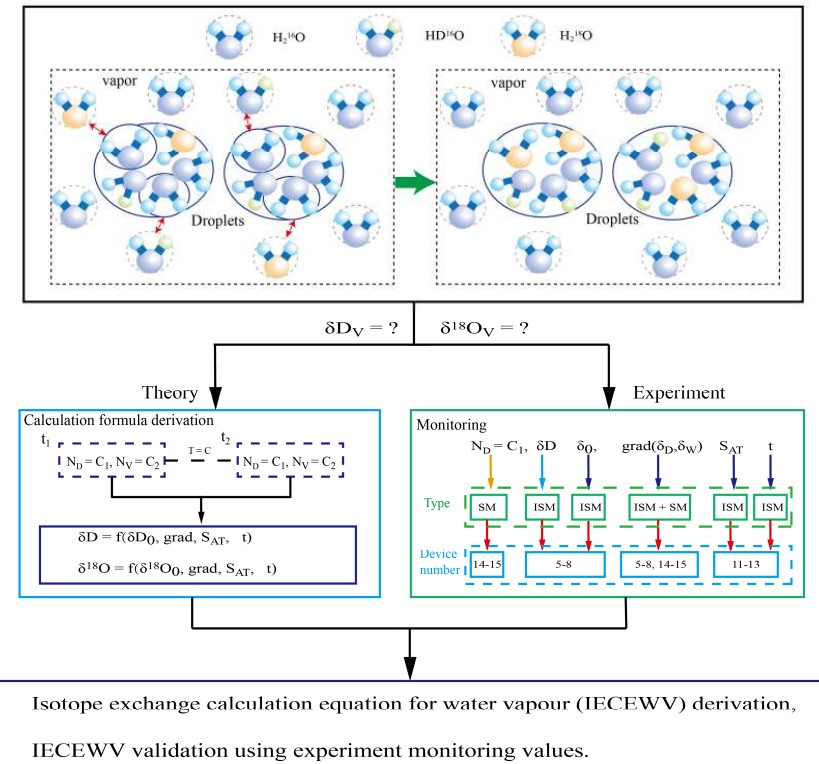

**Figure 1: Correlations between theory and experiment of the article (SM: sampling monitoring; ISM: in-situ**

**monitoring)**



### 2.1 IECEWV

#### 2.1.1 Assumptions

Several assumptions were made for this study.

(1) The exchange was a single-molecule exchange process and there were only three types of molecules in the droplets and

vapour: $H_2O$ (abundant isotopic molecules); HDO (rare isotopic molecules); and $H_2^{18}O$ (rare isotopic molecules).

(2) During the isotopic exchange process between droplets and water vapour, only the exchange between $H_2O$ and $HDO/H_2^{18}O$

was considered and the increase in the rare isotopic molecular number in droplets was equal to the decrease in the rare isotopic

molecular number in water vapour, ensuring the conservation of exchange quantity.

(3) The temperature of the exchange environment was constant and humidity was maintained at 100%.

#### 2.1.2 Derivation process

A single droplet with radius $r$ and total number of molecules $N_D$, where the initial and exchanged isotopic ratios of the droplet

are $R_{D0}$ and $R_{D0}+dR_D$, respectively, and the total number of gaseous water molecules in the water vapour is $N_V$, where the initial

and exchanged isotopic ratios of the water vapour are $R_{v0}$ and $R_{v0}-dR_v$, respectively. Thus

$$(\frac{R_{D0}+dR_D}{1+R_{D0}+dR_D} - \frac{R_{D0}}{1+R_{D0}})N_D = (\frac{R_{V0}-dR_V}{1+R_{V0}-dR_V} - \frac{R_{V0}}{1+R_{V0}})N_V \tag{1}$$

In natural water bodies, the ratios of rare isotopes are typically very small (much less than 1. Therefore, we can simplify the

denominator in Equation (1). This leads to the following:

$$\frac{dR_V}{dR_D} = -\frac{N_D}{N_V} \tag{2}$$

By rearranging Equation (2), we can obtain

$$\frac{dR_V}{dt} = -\frac{N_D}{N_V}\frac{dR_D}{dt} \tag{3}$$

Based on the relationship between isotopic ratios and isotopic values, then

$$R_V = (1+\delta_V)R_r \tag{4}$$

Substituting Equation (4) into Equation (3) and applying the derivative rules, we obtain

$$\frac{d\delta_V}{dt} = -\frac{N_D}{N_V}\frac{d\delta_D}{dt} \tag{5}$$

Equation (5) indicates that the variation in the water vapour isotopes during the exchange process is constrained by the isotopic

variation of the droplet. Referring to reference (Bai et al., 2021b) for the relationships satisfied during the exchange process

between the droplets and water vapour, we have

$$\frac{d\delta^2H_D}{dt} = \frac{18}{19}\frac{1}{m}4\pi r\rho_s\left(\delta^2H_D - \delta^2H_V\right)\lambda\left(\beta_H\right)^\gamma \tag{6}$$


$$\frac{d\delta^{18}O_D}{dt} = \frac{18}{20}\frac{1}{m}4\pi r \rho_s \left(\delta^{18}O_D - \delta^{18}O_V\right)\lambda\left(\beta_O\right)^\gamma \tag{7}$$

where $\rho_s$ is the density of saturated water vapour, $\delta^{18}O_D$ and $\delta^{18}O_V$ are the isotopic values of droplets and water vapour for $^{18}O$,

$\delta^2 H_D$ and $\delta^2 H_V$ are the isotopic values of droplets and water vapour for $^2H$, $\beta_H$ and $\beta_O$ are the diffusion coefficients for HDO

and $H_2^{18}O$ molecules, respectively, $\gamma$ is related to droplet size, and $m$ is the mass of the droplet and its relationship with $N_D$ is

given by

$$N_D = \frac{m}{M_W}N_A \tag{8}$$

where $M_W$ is the molar mass of the droplet (water) and $N_A$ is Avogadro's constant.

By substituting Equations (6) and (7) into Equation (5), we obtain

$$\frac{d\delta^2 H_V}{dt} = -\frac{72}{19}\frac{N_D}{N_V}\frac{1}{m}\pi r \rho_s \left(\delta^2 H_D - \delta^2 H_V\right)\lambda\left(\beta_H\right)^\gamma \tag{9}$$

$$\frac{d\delta^{18}O_V}{dt} = -\frac{72}{20}\frac{N_D}{N_V}\frac{1}{m}\pi r \rho_s \left(\delta^{18}O_D - \delta^{18}O_V\right)\lambda\left(\beta_O\right)^\gamma \tag{10}$$

Equations (9) and (10) represent the relationship between the isotopic values of water vapour and droplets under isothermal

saturated water vapour conditions. By integrating over time and applying the boundary conditions, the isotopic values of water

vapour can be calculated as follows:

$$\delta^2 H_V = \delta^2 H_{V0} - \frac{72}{19}\frac{N_D}{N_V}\frac{1}{m}\pi \rho_s \left(\delta^2 H_D - \delta^2 H_V\right)\lambda\left(\beta_H\right)^\gamma \int r dt \tag{11}$$

$$\delta^{18}O_V = \delta^{18}O_{V0} - \frac{72}{20}\frac{N_D}{N_V}\frac{1}{m}\pi \rho_s \left(\delta^{18}O_D - \delta^{18}O_V\right)\lambda\left(\beta_O\right)^\gamma \int r dt \tag{12}$$

Assuming that the droplet has a regular spherical shape during the falling process, according to the definition of the specific

surface area, we obtain

$$S_{AT} = \frac{3}{r} \tag{13}$$

where $S_{AT}$ is the specific surface area.

In the previous derivation, only the exchange process between a single droplet and water vapour was considered. In reality, an

exchange process occurs between the droplets and the water vapour. Therefore, we extended the above formulas to the

exchange process between the entire droplet and water vapour and use the specific surface area of the droplet instead of the

characteristic droplet diameter as follows:

$$\delta^2 H_V = \delta^2 H_{V0} - \frac{216}{19}\frac{N_D}{N_V}\frac{1}{m}\pi \rho_s \left(\delta^2 H_D - \delta^2 H_V\right)\lambda\left(\beta_H\right)^\gamma \int \frac{1}{S_{AT}} dt \tag{14}$$


$$\delta^{18}O_V = \delta^{18}O_{V0} - \frac{216}{20}\frac{N_D}{N_V}\frac{1}{m}\pi\rho_s\left(\delta^{18}O_D - \delta^{18}O_V\right)\lambda\left(\beta_O\right)^\gamma\int\frac{1}{S_{AT}}dt \tag{15}$$

Equations (14) and (15) show the IECEWV during the exchange process between the droplet and water vapour under

isothermally saturated conditions. Equations (14) and (15) assume that the isotopic values of droplets of different sizes are the

same at the initial moment.

### 2.1.3 Theoretical analysis of IECEWV

The IECEWV revealed that multiple factors collectively influenced the exchange of water vapour isotopes. These factors

include the initial isotope values, molecular number ratio of exchange droplets to water vapour, isotopic gradient, and specific

surface area. Holding other variables constant, a higher molecular number ratio of droplets to water vapour implies a more

intense exchange, which leads to a faster depletion of water vapour isotopes. Similarly, a greater isotope gradient indicates a

stronger force for exchange, resulting in a more significant reduction in water vapour isotope values. Moreover, a faster

reduction in the specific surface area of the droplets indicated a higher rate of droplet coalescence, which resulted in a

significant decrease in water vapour isotopes. It is important to note that the derived computational equation is applicable only

when equilibrium exchange has not been reached and should not be misconstrued as a definition of isotope exchange. The

exchange between droplets and water vapour isotopes also depends on whether the exchange state is in equilibrium, which

will be discussed in sections 3.2.

Next, we analysed the impact of isotope exchange on the linear relationship between hydrogen and oxygen isotopes. We assume

that during the exchange process, the initial hydrogen and oxygen isotope values of both the water vapour and its droplets

adhere to the global precipitation line equation as follows:

$$\delta^2H_{D0} = 8\delta^{18}O_{D0} + 10 \tag{16}$$

$$\delta^2H_{V0} = 8\delta^{18}O_{V0} + 10 \tag{17}$$

By substituting Equations (16) and (17) into Equation (14) and combining them with Equation (15), we obtain

$$\delta^2H_V = 8\delta^{18}O_{V0} - \frac{2800}{323}\frac{216}{20}\frac{N_D}{N_V}\frac{1}{m}\pi\rho_s\left(\delta^{18}O_D - \delta^{18}O_V\right)\lambda\left(\beta_O\right)^\gamma\int\frac{1}{S_{AT}}dt + 10 \approx 8\delta^{18}O_V + 10 \tag{18}$$

Equation (18) indicates that under the conditions of isotope exchange, in terms of hydrogen and oxygen isotopes, the linear

relationship for both water vapour and droplets remains unchanged. When only the slopes of the linear relationships were

identical, the linear relationship of water vapour after exchange was determined by both the initial isotope values and the

difference in slopes. Conversely, if the intercepts of the linear relationships are the same, the linear relationship of the post-

exchange water vapour does not exhibit a specific pattern. When both the droplet and water vapour hydrogen and oxygen

isotope lines passed through the origin but had different slopes, the post-exchange water vapour hydrogen and oxygen isotope



185 lines did not pass through the origin. This conclusion also holds for the post-exchange droplet isotopes. This suggests that the

deviation of precipitation isotope lines from the origin is not solely due to dynamic fractionation processes but also involves

isotope exchange processes.

Based on Equations (14) and (15), in conjunction with the definition of the isotopic excess ($d$-excess), we obtain

$$d = \delta^2 H_V - 8\delta^{18}O_V = (\delta^2 H_{V0} - 8\delta^{18}O_{V0}) + 216 \frac{N_D}{N_V} \frac{1}{m} \pi \rho_s \int \frac{1}{S_{AT}} dt \left[ \frac{1}{20}\left(\delta^{18}O_D - \delta^{18}O_V\right)\lambda\left(\beta_O\right)^\gamma - \frac{1}{19}\left(\delta^2 H_D - \delta^2 H_V\right)\lambda\left(\beta_H\right)^\gamma \right] \quad (19)$$

190 Assuming that the adjustment of isotopic gradients between water vapour and liquid droplets during isotope exchange takes a

long time, it remains essentially constant in the short term. Equation (19) indicates that the post-exchange $d$-excess of water

vapour is primarily influenced by the initial isotopic value of water vapour (the first term on the right side of Equation (19))

as well as the product of the isotopic gradient and the integration term of the reciprocal of the specific surface area (the second

term on the right side of Equation (19)). In this study, the second term is relatively small compared with the first term and can

195 be largely disregarded. Therefore, using this equation for the calculation results in a nearly horizontal line for the $d$-excess of

water vapour, indicating that the $d$-excess remained relatively constant during the isotope exchange process.

When the molecular weight of liquid droplets ($N_D$) is very small, the isotopic equation is simplified to

$$\lim_{N_D \to 0} \delta^2 H_V = \delta^2 H_{V0} \quad (20)$$

The isotopic value of water vapour was equal to the initial isotopic value of water vapour and aligned with the isotopic

200 characteristics of water vapour under isothermal saturation conditions.

## 2.2 Experimental platform

### 2.2.1 Experimental setup

A schematic and physical representation of the experimental setup is shown in Figure 2. The experimental setup consisted of

a droplet generation system (I), exchange chamber (II), online particle size monitoring, temperature, and humidity

205 measurement (III), and isotopic monitoring device (sampling and test) (IV).

The droplet generation system consisted of a pressurised water pump, an outlet, and a pipeline. It operates based on the

principle of physical collisions to generate droplets with different size distributions. The resulting droplets exhibit a bimodal

distribution with a particle size range of 4-500 μm, allowing it to characterise cloud water particle size and its main features

(Song et al., 2018).

210 The exchange chamber consisted of an organic glass tube and a conical bottom plate. The organic glass tube was 2000 mm

high, with a diameter of 600 mm, and nested in the circular groove of the conical bottom plate, which had a slope of 1:2. To

minimise the adhesion of the liquid droplets to the interior walls of the exchange chamber, the entire inner surface was coated

with a superhydrophobic coating. The top of the exchange chamber is covered with a plastic film isolated from the external





environment. After exchanging the droplets with water vapour in the chamber, the droplets were collected through a conical
bottom plate and flowed into a beaker.

An online particle size monitoring system includes a laser particle size analyser and temperature and humidity sensors. A laser
particle size analyser was used to monitor the droplet size distribution at the bottom of the exchange chamber. The observation
window was located 700 mm above the bottom of the organic glass tube and had a circular hole with a diameter of 100 mm
along the radial direction of the exchange chamber, allowing the observation of the laser beam. The laser particle size analyser,
Winner 319 (Jinan Veina), is a separate laser particle size analyser with a measurement range of 1–500 μm, time resolution of
1 s, and observation error ≤ 3 %. Temperature and humidity sensors were placed in the middle of the exchange chamber to
provide real-time monitoring of the temperature and humidity. The temperature and humidity measurement accuracy is ± 1 ℃
and ± 1 %, respectively.

The isotopic monitoring device consisted of a water vapour isotope monitor (ABB, GLA431-TIWA), a droplet sampling bottle,
a collection beaker, and a weighing device. The water vapour isotope monitoring port was positioned 150 mm from the top of
the exchange chamber. The collected water vapour was transported via a pipeline to an isotope analyser for real-time
measurements. The time resolution of the measurement was 1 s and the analysis precision for $\delta^{18}$O and $\delta^2$H were 0.05 ‰ and
0.2 ‰, respectively. A beaker and droplet sampling bottle were used to collect the settled and post-settled liquid water in the
exchange chamber, respectively. After collection, the beaker and sampling bottle were weighed and the samples were retained
for testing.

To simulate the environment of isotopic exchange between the droplets and water vapour during droplet size enlargement, a
noncontact acoustic intervention device was added. Low-frequency sound waves can promote droplet enlargement (Bai et al.,
2021a; Bai et al., 2020), simulating the early stages of warm cloud transformation into rainfall, specifically the processes of
droplet collision and coalescence to form raindrops. This study investigated the isotopic changes in water vapour during droplet
growth. The impact of noncontact intervention on the diffusion coefficients of water molecules in droplets and water vapour
was not addressed in this study.

### 2.2.2 Experimental process and conditions

Before the experiment, the laser particle size analyser was calibrated. Once the measurement error and repeatability error were
both below 3%, indicating that the precision of the measurements met the requirements and the observed data were stable, the
laser particle size analyser was adjusted to the observation state. The exchange chamber continuously introduced droplets,
whereas the isotope analyser and temperature-humidity sensor began continuous observations. When the relative humidity of
water vapour in the exchange chamber reached saturation (100 %), the droplet size was recorded, and the collection of droplet
samples began. The collection was performed for 300 s. Subsequently, the acoustic intervention device was activated, and the


droplet sampling bottle and beaker were replaced within 2 s. The time was recorded and sample collection continued for

another 300 s. The noncontact intervention device was then turned off, and the entire experimental process was repeated. The

exchange chamber temperature is maintained at the ambient temperature of 20 °C.

The observation section implements preset corrections for concentration and time dependence (Galewsky et al., 2016) to reduce

the impact of concentration and time on the accuracy of isotopic observations. Concentration dependence was corrected by

setting different concentrations of standard water vapour sources, and the actual observed values were corrected for

concentrations based on the test results. Time dependence was corrected using recognised standard water vapour sources.

Standard water vapour samples were examined for concentration and time drift correction every ten hours during the

experimental procedure. Time correction was started initially and a concentration-dependent correction was performed after

ten hours of observation.

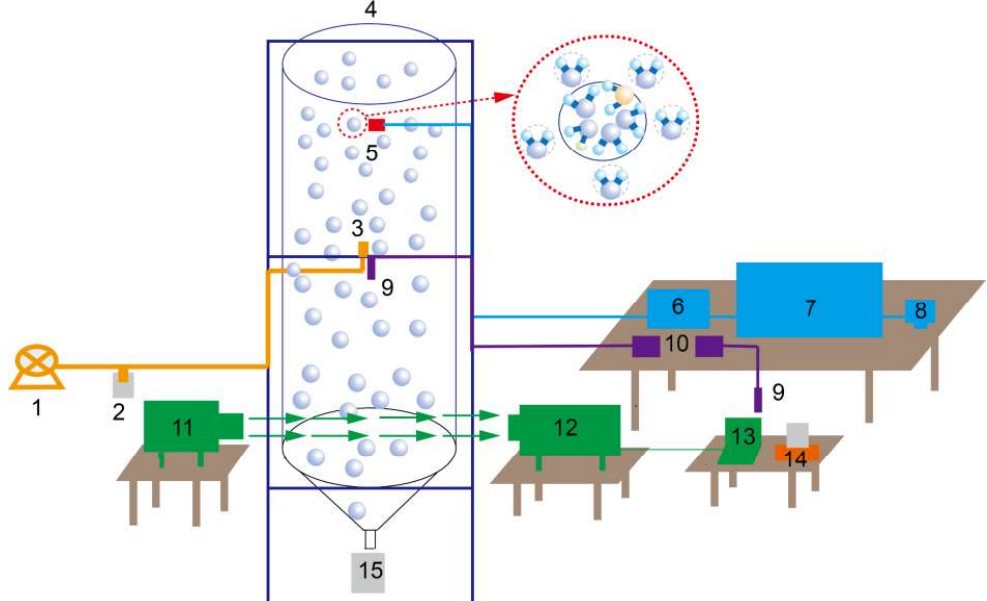



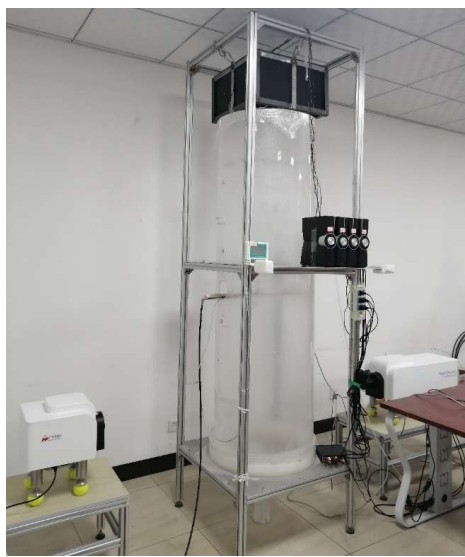
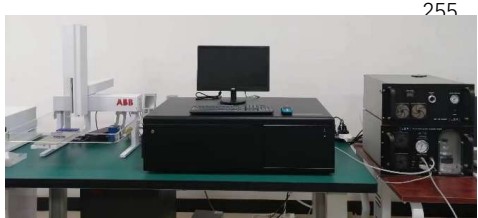
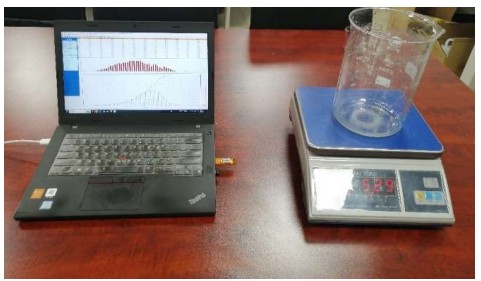

**Figure 2: Experimental Setup and Actual Configuration** (1- Water pump, 2- Droplet sampling bottle, 3- Droplet outlet, 4- Exchange chamber and bracket, 5- Water vapour isotope sampling port, 6- Water vapour isotope standard sample generator, 7- Isotope analyzer, 8- Isotope display, 9- Temperature and humidity probe, 10- Temperature and humidity display panel, 11- Laser emission end, 12- Laser reception end, 13- Droplet size analyzer computer, 14- Weighing device, 15- Beaker; I includes 1 and 3, II is exchange chamber 4, III includes 9-13, IV includes 2, 5-8 and 14-15.)

Sample preparation was finished within a day after each sample collection was finished, and the sample storage bottle was purged of extra air and kept in a refrigerator at 4 °C. Over 60,000 data points for isotopes and droplet particle sizes were produced during the procedure, including 400 liquid water samples.

**2.2.3 Data processing for the experiment**

(1) Droplet size data processing: Droplet size spectrum data collected from 100 control groups (acoustic intervention and natural conditions) were individually time-averaged to characterise the initial particle size and droplet size spectrum after enlargement. Additionally, to elucidate the process of particle size variation, the characteristic particle size was averaged over experimental runs, resulting in temporal data for the variation in $D_{90}$ (This means that the volume frequency of droplets smaller than this particle size accounts for 90% of the total volume) over time. Similarly, the specific surface area was obtained over time.

(2) Isotope data: To eliminate observational errors, the water vapour isotope data underwent concentration and time drift corrections. The water vapour isotope data were averaged using the same approach as that used for the aforementioned characteristic particle size. Unlike the particle size data observed by the laser particle size analyser, the observed water vapour



passed through a certain length of tubing, causing a delay in the actual time of isotope observation. However, the delay time was constant (60 s; the speed of water vapour in the tubing remained constant). By observing the delayed time of isotope changes between the observed tubing water vapour values and standard water vapour values, the time of observation for water vapour isotopes was corrected and the actual time-dependent water vapour isotope was obtained. To verify the isotope exchange process, liquid water was used to analyse the isotopic changes before and after exchange, and the liquid water weight

was used to verify the conservation of the droplet mass after exchange (Bai et al., 2021a).

## 3 Experiment results and analysis

### 3.1 Variations of droplets microphysical parameters and temporal evolution characteristics of water vapour isotopes

Figure 3(a) shows the cumulative volume frequency change in the observed droplet sizes. This shows that all distinctive droplet sizes increased after noncontact acoustic intervention, whereas the droplet peak form remained intact. The main peak value of

the droplet size shifts to the right, and the size distribution widens. In Figure 3(a), under natural conditions, the size distribution had a width of 145 µm, with the main peak at 15 µm. After non-contact ultrasonic intervention, the size distribution widened by 28 µm, and the main peak shifted to 38 µm.

Figure 3(b) illustrates the time evolution of the characteristic droplet size ($D_{90}$) and specific surface area ($S_{AT}$) with increasing intervention time. Equilibrium was reached at 109 s, with an initial increase/decrease followed by stabilisation. The droplets

primarily settled after entering the exchange chamber. From the observed changes in droplet size over time, it can be deduced that the droplets entered the exchange chamber and fell to the conical bottom plate within 109 s on average, indicating an isotopic exchange time for droplets to water vapour. When other factors remained constant, the specific surface area effectively measured the potential of the droplets to exchange energy and substances with the surrounding medium. A larger specific surface area indicates a greater potential for the exchange contact area, when the saturated exchange state was not reached, the

potential for isotopic exchange between the droplets and water vapour was higher.

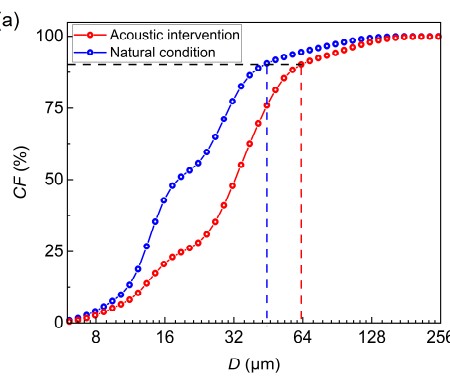
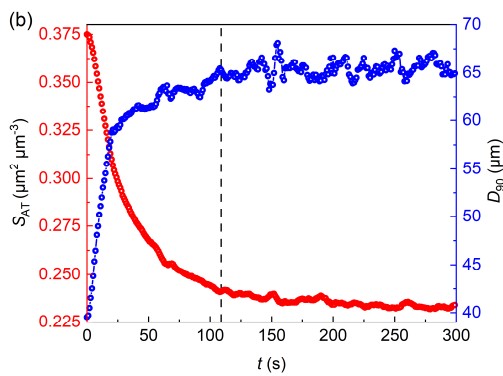



**Figure 3: Cumulative volume frequency of droplets (a) and temporal variations of characteristic droplet size $D_{90}$ and specific**

**surface area $S_{AT}$ (b)**

Figure 4 illustrates the observed alterations in the water vapour isotopes within the exchange chamber. Specifically, $\delta^2H$ and $\delta^{18}O$ of water vapour show an initial increase, a subsequent decrease, and a final stabilisation. Water vapour isotope variations within the exchange chamber can be divided into three stages based on changes in the microphysical properties of the droplets and their settling periods (one settling cycle takes 109 s). From 0 to 109 s, both $\delta^2H$ and $\delta^{18}O$ show an enrichment trend, with values increasing from -83.88 ‰ and -13.33 ‰ to -83.75 ‰ and -13.32 ‰, respectively. From 109 to 218 s, water vapour $\delta^2H$ and $\delta^{18}O$ drop to -84.01‰ and -13.36‰, and then keep comparatively steady from 218 to 300 s. The changes in $\delta^2H$ and $\delta^{18}O$ values exhibit good consistency, with the hydrogen isotope changes showing a greater magnitude than the oxygen isotope changes influenced by molecular activity. The observed characteristics of the water vapour isotopes further confirmed the accuracy of the estimated average settling time of the droplets based on microphysical features.

The changes in the water vapour isotopes are attributed to the isotopic exchange between the droplets and water vapour (Bai et al., 2022b; Bai et al., 2021b). The water pump slightly increased the temperature of the droplets entering the exchange chamber during the experiment. Nonetheless, the rate of temperature rise was low at $1.6 \times 10^{-4}$ °C s$^{-1}$, meaning that an average temperature increase of 0.017 °C occurred during a single settling cycle. Consequently, the effect of temperature variation on the molecular quantity of water vapour in the exchange chamber can be disregarded. The exchange chamber's relative humidity remained constant at 100 % (± 0.1 %) during the experiment, indicating that the water vapour in the exchange chamber was saturated. Therefore, it is reasonable to ignore how temperature and humidity variations in the chamber and surrounding air affect the isotopic values of water vapour (even if there is an impact, it would cause a monotonic change in isotopic values). Thus, the changes in the water vapour isotopes were primarily influenced by the isotopic exchange between the droplets and water vapour. Furthermore, based on the mass conservation of droplets during the settling process, combined with the isotopic increment before and after settling, and assuming a constant input of droplets into the exchange chamber within a single settling cycle, the exchange rate of droplets with water vapour molecules was estimated at $2.10 \times 10^{20}$ /s.





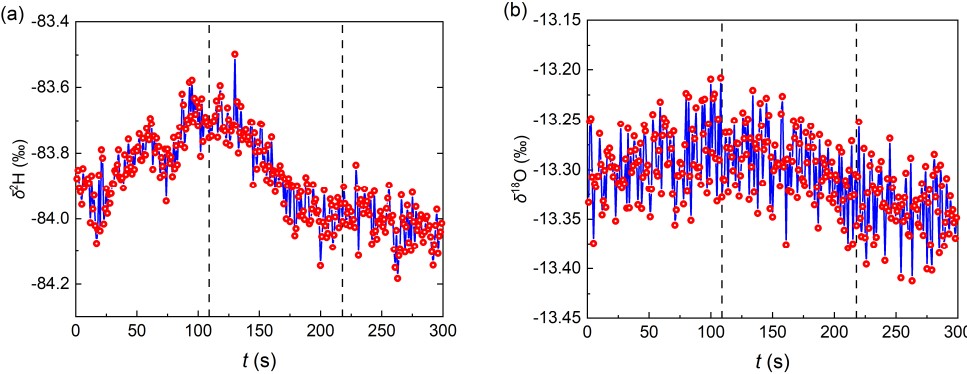

**Figure 4: Temporal variation of hydrogen and oxygen isotopes in water vapour within the exchange chamber: $\delta^2H$ (a) and $\delta^{18}O$ (b)**

### 3.2 Analysis of factors influencing isotopic variations in water vapour

The variation in the isotopic values of water vapour is influenced by a combination of factors, such as exchange state, droplet-specific surface area and isotopic gradients, which results in an initial enrichment, followed by depletion, eventually reaching a stable state of isotopic values. This phenomenon is attributed to the dynamic equilibrium between aggregation and fragmentation in the exchange chamber (Bai et al., 2022a; Shi et al., 2020). Before the droplet size increases (prior to acoustic intervention), the exchange of isotopes between the droplets and water vapour reaches equilibrium (Bai et al., 2021b), this implies that the gradient of rare isotope molecules on the surface of the liquid droplets and in the water vapour reached an exchange equilibrium. Despite the existence of isotopic gradient differences, the isotopic gradients between the liquid phase and the gas phase reached a balanced state because of the phase differences, although exchange still occurs, there is no longer a change in isotopic values between liquid droplets and water vapour.

As the droplet size increases, the degree of rare isotope distribution on the surface of the liquid droplets increases, disrupting the previously established equilibrium exchange state. This disruption initially results in a reversal of the exchange direction, leading to an increase in water vapour isotopic values. During the period from 0 to 109 s, the increase in droplet size caused by droplet aggregation enlarged the droplet size and reduced the particle number, which enhanced the degree of rare isotope distribution on the surface of the liquid droplets, causing isotopic exchange to progress in the direction of depleting isotopic values in liquid droplets and enriching isotopic values in water vapour. Because of the larger molecular mass and faster diffusion speed of water molecules in the exchange chamber, the isotopic exchange induced by changes in the degree of rare isotope distribution reached equilibrium within a settling period (109 s), as observed in Figure 4 (a), and there was a brief stage of stable water vapour isotopic values within the time range of 107 to 113 s. Subsequently, isotopic exchange was influenced by the specific surface area.

Influenced by the specific surface area of the droplets, the isotopic values of water vapour exhibited an initial depletion trend,





followed by stabilisation from 109 to 300 s. As the droplets entered the exchange chamber and settled (109–218 s), the specific surface area of the droplets decreased, as shown in Figure 3 (b). During this phase, isotopic exchange between the droplets and

water vapour led to the continuous enrichment of isotopes in the droplets (Bai et al., 2021b), resulting in the depletion of isotopes in the water vapour. The rate of isotopic depletion decreased over time owing to a reduction in the rate of change of the specific surface area (as the specific surface area decreases with time, attributed to the increase in droplet size, decrease in particle number, and consequent increase in the average free path of gaseous molecules in water vapour colliding with the surface of liquid droplets. This decrease in the probability of liquid droplets capturing water vapour led to a reduction in the

exchange amount, causing a decrease in the exchange rate coefficient). From 219 to 300 s, the isotopic exchange reached a new equilibrium state, with the water vapour isotopic values in dynamic equilibrium.

## 4 Discussion

### 4.1 Experimental and theoretical comparison analysis of stable isotopic values of water vapour and their statistical error characteristics

The calculated values effectively captured the variations in the water vapour isotopes. Utilizing the derived IECEWV as described in the article, calculations were performed on the water vapour isotopes. The calculated values were averaged based on the number of experimental sets, and the calculated and observed water vapour isotopes during the 109–218 s period were compared (Figure 5). The calculated values of water vapour $\delta^2H$ and $\delta^{18}O$ decreased over time, exhibiting a trend consistent with the observed values. The calculated values at 109s matched the observed values, this was attributed to the use of actual

observations for the initial values. Additionally, at 218 s, the calculated values for water vapour $\delta^2H$ and $\delta^{18}O$ were -84.01 ‰ and -13.36 ‰, respectively, while the observed values were -84.00 ‰ and -13.35 ‰. The agreement between the calculated and observed values indicates that IECEWV effectively described the exchange process between the droplets and water vapour.

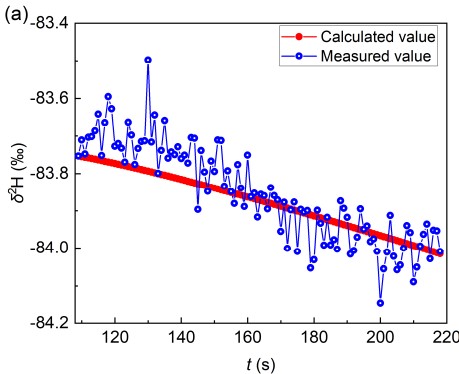
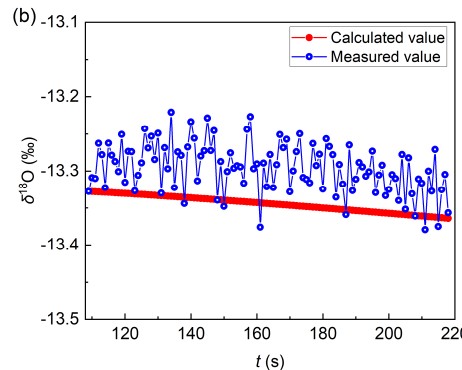

**Figure 5: Calculated and observed water vapour isotopes during the droplets-water vapour exchange process: $\delta^2H$ (a) and $\delta^{18}O$ (b)**





The mean relative errors of water vapour $\delta^2H$ and $\delta^{18}O$ showed a slightly increasing trend over time but there was no significant cumulative error. Averaging the relative errors for 95 sets of calculations, the mean relative errors and cumulative errors (obtained by accumulating the relative errors) for $\delta^2H$ and $\delta^{18}O$ from 109 to 218 s are shown in Figure 6. The mean relative errors for $\delta^2H$ and $\delta^{18}O$ showed a certain growth trend with increasing calculation time but the increment was not substantial. The relative error for $\delta^2H$ increased from 0.74% to 1.66% and for $\delta^{18}O$ from 2.77% to 3.19%. The mean relative errors for $\delta^2H$

and $\delta^{18}O$ increased by 0.92% and 0.42%, respectively, with relative error time increments of $8.5\times10^{-3}$ % s$^{-1}$ and $3.9\times10^{-3}$ % s$^{-1}$. If the calculation method provided in this study accumulated errors over time, the relative errors for $\delta^2H$ and $\delta^{18}O$ would reach 124 % and 321 %, respectively, by the end of the calculation period. However, the relative errors at the end of the calculation period were only 1.3 % and 1.0 % of the cumulative errors, respectively. Therefore, the derived equation for water vapour isotope exchange presented in this study did not result in significant cumulative errors over time.

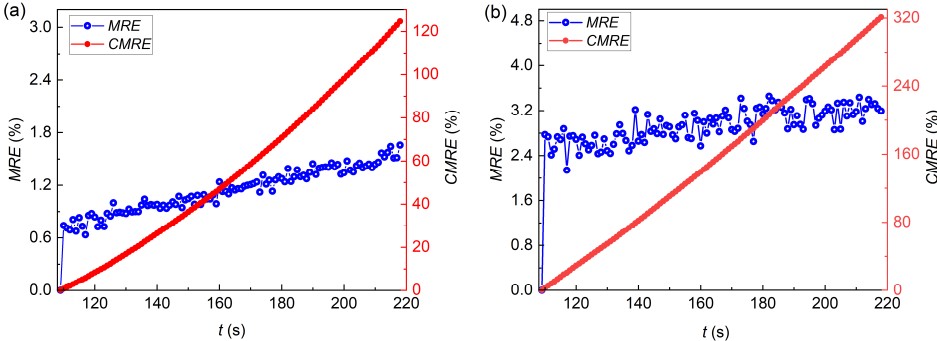

**Figure 6: Temporal variation of mean relative errors and cumulative mean relative errors for water vapour $\delta^2H$ (a) and $\delta^{18}O$ (b) calculations**

The cumulative frequencies of relative errors for $\delta^2H$ and $\delta^{18}O$ were over 98 % when the relative errors were less than 9.5 % and the majority of the calculation errors were concentrated within 4.5 %. We analysed the frequency distribution of 10,355

calculated relative errors (Figure 7), and found that the relative errors for both $\delta^2H$ and $\delta^{18}O$ were primarily within 4.5%, with cumulative frequencies reaching 98.8% and 82.8%, respectively. The frequency decreased with increasing relative errors and the rate of decrease was more pronounced for $\delta^2H$ than for $\delta^{18}O$, consistent with Figure 5(b). Although the trend of calculated $\delta^{18}O$ values matched the observed trend, the calculated values closely aligned with the lower limit of the observed values. The relative error for $\delta^2H$ was lower than that for $\delta^{18}O$ due to the baseline values.



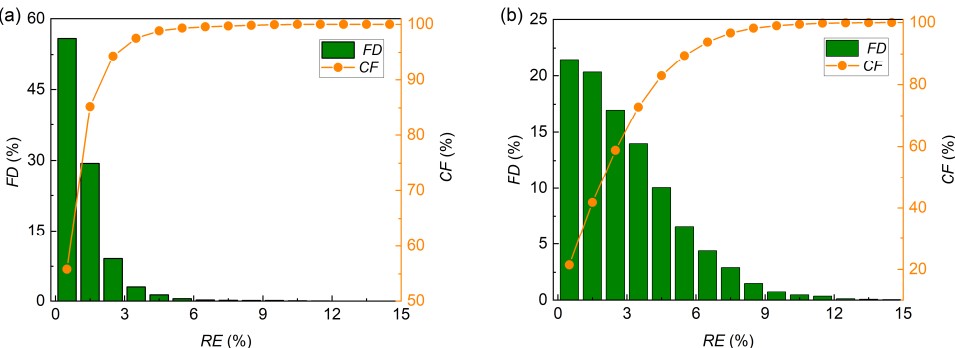


**Figure 7: Frequency distribution and cumulative frequency distribution of relative errors of water vapour $\delta^2$H (a) and**

**$\delta^{18}$O (b) calculations**

**4.2 Linear relationship between $\delta^2$H and $\delta^{18}$O during the isotope exchange process, along with its $d$-excess characteristics**

Under saturated water vapour conditions, $d$-excess remained constant during the isotopic exchange between the droplets and water vapour. Averaging 95 sets of isotope calculations and observations for the 109–218 s period, a graph depicting the variation in $d$-excess over time was obtained (Figure 8(a)). The observed $d$-excess values exhibited fluctuating changes over time, with a mean of 22.52 ‰. According to the IECEW, the calculated $d$-excess remained essentially a horizontal line with a mean of 22.88 ‰. The calculated $d$-excess values were slightly larger than the observed values, attributed to the calculated

$\delta^{18}$O being smaller than the observed values.

The slope of the linear relationship can be characterised by the distribution of clusters with the same slope. A linear relationship fit to the calculated hydrogen and oxygen isotope values revealed that during the exchange process, the isotope linear relationship was $\delta^2$H=7.19$\delta^{18}$O+12.14 ($n$=95, $R^2$=1.0, significance level 0.05). The slope was slightly smaller than that of the global precipitation line but the intercept was slightly larger. The experimental observations of water vapour isotopes exhibited

fluctuations, resulting in a significant difference between the linear relationship fit to the observed data and that of the calculated values. Therefore, adopting the cluster fitting approach, a linear cluster with the same slope was formed (Figure 8(b)), and the experimental points were well distributed around the linear cluster (four lines with a slope of 7.19, and intercepts for the horizontal axis from left to right were -13.392, -13.365, -13.338, and-13.305, with a bandwidth of 0.027). This suggests that under saturated water vapour conditions, the linear relationship of isotopes can be effectively represented by a linear

cluster with the same slope, although further refined experiments are required for validation.



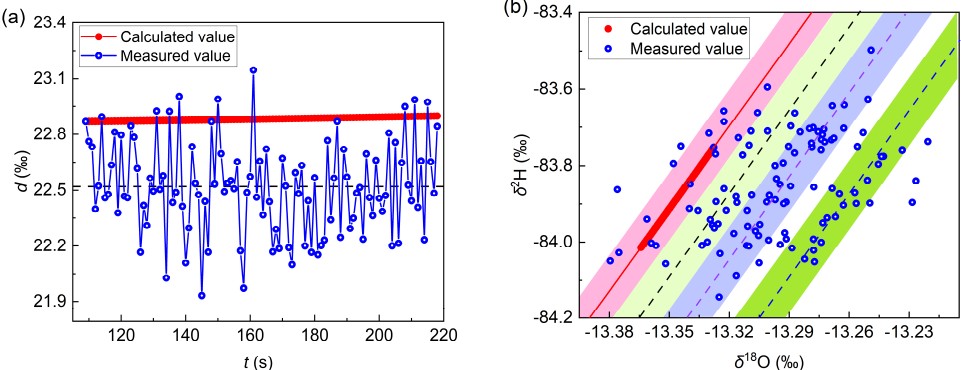

**Figure 8: Characteristics of *d*-excess variation (a) and its hydrogen and oxygen isotope relationship (b)**

### 4.3 Analysis of uncertainty factors in the experiment and IECEWV

Uncertainty was caused by the values of the hydrogen and oxygen isotope molecular diffusion coefficients. In this study, the values of the hydrogen and oxygen isotope molecular diffusion coefficients in the water vapour isotope calculation equation were referenced from the values under typical air conditions, which may introduce certain errors. This was because the exchange chamber was in a saturated state with the air moisture reaching its maximum. Therefore, the values of hydrogen and oxygen isotopes in liquid droplets differed from those under general experimental conditions and tended to be somewhat larger, resulting in a certain degree of underestimation in the calculated values. However, there are currently no measured data available for reference under saturated water-vapour conditions.

Uncertainty also arose from the use of the specific surface area of the droplets instead of the droplet size. The variation in water vapour isotopes is influenced by the microphysical characteristics of the exchange object, that is, the droplets. In the derivation process, the specific surface area of the droplets was used instead of the actual droplet particle size for the calculations. When using droplet particle size for calculations, it is necessary to consider the number of droplets with the same particle size and their volume proportions within the droplets. Isotopic values are then given based on the volume mixing ratio, and it is also necessary to consider the settling time of different sizes in the exchange chamber. Therefore, the use of the specific surface area of a generalised droplet for calculations leads to a certain degree of underestimation of the calculated values.

The lack of real-time updating of droplet-water vapour isotope gradients also led to uncertainty. During the experimental process, droplet aggregation can lead to changes in the distribution of rare isotope molecules on the droplet surface. In addition, the droplets were not perfect spheres during settling and there were no observed data on the shape-change process after aggregation. Droplet aggregation can lead to changes in isotopic gradients during the exchange process (gradients are the driving factors for isotopic exchange), causing fluctuations in water vapour isotopes, as confirmed by the observed fluctuations in hydrogen and oxygen isotopes (Figure 5). In summary, the changes in the isotopic gradients between droplets and water

vapour due to droplet aggregation are complex and neither experimental nor calculation conditions can account for them. This

complexity is a significant reason why the calculated values do not closely match the observed values, especially for fluctuation

processes.

In actual precipitation processes, the variation in water vapour isotopes is more complex, but the study provided theoretical

calculation formulas and characteristics of water vapour isotopic changes under constant temperature and saturated conditions.

In the future, water vapour isotopic values under saturated or near-saturated conditions should be calculated without in situ

observations by combining this equation with observed raindrop spectral data. By integrating the Rayleigh distillation model,

this approach holds significant application value for exploring the coordinated changes in precipitation and water vapour

isotopes under cooling and saturated conditions.

## 5 Conclusion

To explore the influence of the exchange process on water vapour isotopes and improve our understanding of the isotope

mechanism of the precipitation process, this study derived the IECEWV under isothermal saturation conditions based on the

conservation of mass during the exchange process between liquid droplets and water vapour isotopes. Additionally, indoor

experiments were designed and conducted for observational purposes and the calculated values were compared and validated

against the observed values. An analysis of the uncertainties in the experimental and calculation equations was performed,

leading to the following conclusions:

(1) Based on the principle of equal exchange quantities during the exchange process, we derived the IECEWV during the

exchange process between droplets and water vapour under isothermal saturation conditions. Theoretical analysis indicated

that the isotopic values of water vapour were influenced by the initial isotope values, molecular number ratio of exchange

droplets to water vapour, isotopic gradient, and specific surface area of the droplets. The isotope exchange process can lead to

the deviation of hydrogen and oxygen isotope lines from the origin (similar to the dynamic fractionation process) and can be

applied to both water vapour and droplets. It is important to note that the IECEWV is applicable only when an equilibrium

exchange has not been reached.

(2) The exchange process was affected by the degree of rare isotope distribution on the surface of the liquid droplets and the

exchange state. When the characteristic particle size of the droplets first increased and then stabilised, the isotopic values of

water vapour exhibited a trend of initial enrichment, followed by depletion, and then stabilisation. The trend in the calculated

isotopic values of water vapour aligned with the trend observed in the experiments. The average maximum relative errors for

$\delta^2$H and $\delta^{18}$O calculations were 1.66 % and 3.19 %, respectively. The relative errors were primarily concentrated within 4.5 %,

with a cumulative frequency exceeding 98 % for relative errors less than 9.5 %.

(3) The linear relationship between hydrogen and oxygen isotopes in water vapour during the exchange process can be characterised by linear clusters with the same slope under saturated water vapour conditions. The $d$-excess remained constant

during the isotopic exchange between the droplets and water vapour, serving as a criterion for determining whether an exchange process has occurred.

(4) We did not observe the molecular distribution process during droplet aggregation. In the future, we plan to construct a more refined observational platform for further investigation. For real-world scenarios involving cooling saturation leading to rainfall, the combined application of the IECEWV and Rayleigh fractionation models can be valuable as it has significant

applications in exploring coordinated changes in precipitation and atmospheric water vapour isotopes.

## Declaration of Competing Interest

The authors declare that they have no conflict of interest.

## Author contributions

Wenwen Bai: Conceptualisation, methodology, and writing-original draft preparation; Jiahua Wei: Supervision, methodology,

writing-review, and editing; Sanchuan Ni: Investigation, formal analysis, writing-review, and editing; Zhanyu Yao: Funding acquisition; Yifeng Liu, Jingjing Ding, Kaiyu Wang, and Nan Liu: data curation; Miao Wu and Liner Wang: Visualisation.

## Acknowledgments

This study was supported by the National Natural Science Foundation of China (Grant Nos. 52209092 and 42375198),

Natural Science Foundation of Technology Department of Qinghai Province (Grant No. 2023-ZJ-971Q).

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
