# Peer review of "Technical note: Theoretical and experimental investigation of isotopic exchange between water vapour and droplets under isothermal saturation conditions"

_Atmospheric Chemistry and Physics, 2024_

## Referee Comment (RC1)

**Review acp-2024-4**

Technical note: Theoretical and experimental investigation of isotopic exchange between water vapour and droplets under isothermal saturation conditions

Wenwen Bai, Jiahua Wei, Sanchuan Ni, Zhanyu Yao, Yifeng Liu, Jingjing Ding, Kaiyu Wang, Nan Liu, Liner Wang, and Miao Wu

The manuscript "Theoretical and experimental investigation of isotopic exchange between water vapour and droplets under isothermal saturation conditions" introduces an approach to estimate the effect of isotopic exchange between water vapour and droplets in saturated conditions on the isotopic composition of water vapour. The theoretical approach allows to calculate the change in isotopic composition of water vapour during this process from environmental factors. The presented data show a similar trend in the isotopic composition from theory and experiment during a time period of 5min.
This study addresses an important question of the effect of isotopic exchange under saturated conditions between droplets and water vapour on the isotopic composition of water vapour. This effect can, for example, not be modeled by the Craig and Gordon (1965) model due to non-linear effects close to saturated conditions. The authors use an interesting experimental approach to test their model. But in its current state, the manuscript does not convincingly show the relevance of their results due to (i) missing definition of the analysed exchange process and how it differs from established terms such as isotopic equilibration between phases, (ii) missing reference to existing models such as equilibrium fractionation factors and Stewart model for the exchange between droplets and water vapour, and (iii) weak statistical analysis without quantitative uncertainty estimates of their experimental and modeled results. In the following, these three main comments are explained in more detail, together with further comments.

**Main comments:**

- What is the definition of *exchange process*? What is its role for equilibrium fractionation and what is, for example, the difference between equilibration between water vapour and the rain droplets (as e.g. mention on lines 76-77) and the exchange process?

- The experimental results are only compared to the model introduced in this study. It is difficult to relate the results, especially the magnitude of the observed changes in $\delta D$ and $\delta^{18}O$, to the fractionation effect as simulated by other, existing models (e.g. equilibrium fractionation effect, Craig-Gordon (1965) model, Stewart (1975) model). Some of these model cannot be applied at 100% relative humidity but any discussion of these limitations and how the theoretical approach of this study is filling a gap is missing. The study should outline more clearly how their results relate to previous theoretical models and frameworks.

- Statistical analysis: The discussion of the difference between the experiment and the modeled data focuses mainly on the relative difference between the two datasets. This discussion does not address the uncertainty of the experiment. Do the differences fall within the uncertainty and are the changes during the 300s significant with respect to the differences between different model runs? Further, the cluster analysis is not documented and the interpretation of the clusters as a linear relationship between $\delta D$ and $\delta^{18}O$ is questionable.

**Further comments**

- General:
  - The manuscript is often difficult to read due to incomplete sentences, incorrectly used words and grammatical inconsistencies. I do not list all of these issues below, please thoroughly check the language of the manuscript again.
  - The figure captions are often too short. They lack information on the shown data or schematics. Please, improve the captions such that the reader can understand the figures by reading the captions without reading the text.

- Abstract:
  - There are several terms used in the abstract that are not defined/generally known in the field: exchange process (line 17), hydrogen and oxygen isotope line (line 25), the origin (line 25).
  - It is difficult to follow the description of the results in the abstract without having read the manuscript.

- Introduction:
  - Line 62: " in an isotope exchange scenario" do you mean "in an isotope exchange-**only** scenario"
  - Line 67: "Gu et al. 2011" Can you add a reference to earlier work on this topic?
  - Lines 70-71: "kinetic and diffusion fractionation" What is the difference between kinetic and difussion fractionation?
  - Lines 71-72: "In many cases, the Rayleigh fractionation model is utilised to describe such nonequilibrium processes (Liu et al., 2015)"
    - The Rayleigh model was introduced by Dansgaard (1954)
    - The Rayleigh model describes the evolution of the isotopic composition of water vapour and rain during a rain-out process. This can involve non-equilibrium fractionation if the cloud droplets are forming in over-saturated conditions. But I would not use it as typical example of non-equilibrium fractionation.
  - Lines 76-77: " the isotope exchange process leads to weaker equilibration on the water vapor and precipitation isotope values (Graf et al., 2019)" what do you mean by *weaker equilibration*?
  - Lines 85-87: "The study will deeply our understanding of isotopic changes in the water cycle, especially for coordinated changes in precipitation and water vapour isotopes, and further expand the applicable accuracy and scope of the Rayleigh fractionation model." Why are results of this study (especially) relevant for the Rayleigh fractionation model?
  - The discussion of literature stays mostly on a macroscopic level without further discussing the processes on a molecular level even though these processes are important in the methods and results section. How has water vapour – droplet interaction been discussed in literature? How does the approach of this study differ from previous approaches?

- Methods and materials
  - In this section, past and present tenses mix and it is not always clear why the tense changes. Please, check again their usage.
  - Fig.1: The schematic of the experiment (green box) is difficult to understand. What are the numbers? What are *Type*? What do the arrows mean?
  - Equation (1): more explanation is needed how equation 1 is derived. What does it represent? Does follow from the assumption that the rate of change in water molecules in the vapour phase and droplets are equal?
  - Equation (4): What is $R_r$?
  - Equation (6): What is $\gamma$?
  - Line 166: "significant reduction": What is a significant reduction? Did you do a statistical test?
  - Line 166-167: "a faster reduction in the specific surface area of the droplets indicated a higher rate of droplet coalescence, which resulted in a significant decrease in water vapour isotopes."
    Is this still theoretical or does this refer to the experiment?

- ○ Line 192: "it remains essentially constant" What is "it"?
- ○ Fig.2 caption: " I includes 1 and 3, Ⅱ is exchange chamber 4, Ⅲ includes 9-13, Ⅳ includes 2, 5-8 and 14-15)" What are I, II, III and IV?
- ○ Fig. 2: The inlet of the water vapour analysis (5) is above the droplet inlet (3). How do you make sure that the droplets distribute through the entire chamber before they settle?
- ○ Lines 273-274: "Over 60,000 data points for isotopes and droplet particle sizes were
- ○ produced during the procedure, including 400 liquid water samples." Why do you mention these numbers? They do not help to understand the outcome of the experiments. It would, e.g. be more useful to know the number of experiment runs.
- ○ Lines 278-279 and lines 283-284: "The characteristic particle size was averaged over
- ○ experimental runs" and "The water vapour isotope data were averaged using the same approach as that used for the aforementioned characteristic particle size." How was this averaging done? Are all runs averaged to derived mean temporal evolutions of the measured quantities during 300s runs?

- • Experiment results and analysis
  - ○ Line 299: "increase/decrease" What does this mean? Please, be more specific.
  - ○ Lines 303-305: This sentence is too long.
  - ○ Fig. 3-5:  Do you show means over all experimental runs in Fig. 3-5 means? Can you add a standard deviation (or some uncertainty estimate) around the mean value?
  - ○ Fig. 3: What are the vertical dashed lines in (a)?
  - ○ Fig. 3: The changes in  $\delta D$ and $\delta^{18}O$ are relatively small in this experiment. Is this caused by the small difference in the isotopic composition between droplets and vapour? What magnitude of changes do you expect if the isotopic composition differs more strongly between the droplet and its surrounding?
  - ○ Lines 342-343: "As the droplet size increases, the degree of rare isotope distribution on the surface of the liquid droplets increases, disrupting the previously established equilibrium exchange state." Why does the distribution of rare isotopes change with an increase in droplet size? What is the assumption on the isotopic distribution in a droplet? Is it well-mixed? Or is there a gradient between surface and inner part of the droplet?

- • Discussion
  - ○ Lines 365: "The calculated values" Values of what?
  - ○ Lines 366-368: "The calculated values were averaged based on the number of experimental sets, and the calculated and observed water vapour isotopes during the 109–218 s period were compared (Figure 5)." Why is the model only compared to measurements after 109s? Why does the model not apply to the first period?
  - ○ What is the initial value of  $\delta D_D$ and $\delta^{18}O_D$ in the model?
  - ○ Line 375:  "The mean relative errors of water vapour $\delta D$ and $\delta^{18}O$ showed" Between modeled and measured  $\delta D$ and $\delta^{18}O$?
  - ○ Line 378: "not substantial" What do you mean by *not substantial*?
  - ○ Lines 388-394: The paragraph on relative errors and their discussion is difficult to follow and the meaning of the discussion is not clear. Why is a relative error of 4.5% chosen as reference? How does the relative error help to assess the model performance?
  - ○ Line 408: "$R^2=1.0$": What is R? And if it is equal to 1, does this mean that all the data points lie perfectly on a line?
  - ○ Line 411: Cluster fitting approach: What is done here? What is the meaning of the clusters? How does this prove a linear relationship between $\delta D$ and $\delta^{18}O$?
  - ○ Line 445-447: "By integrating the Rayleigh distillation model, this approach holds significant application value for exploring the coordinated changes in precipitation and water vapour isotopes under cooling and saturated conditions." What do you mean by integrating the Rayleigh model? Is the isotope exchange not already part of the Rayleigh model through the isotopic equilibrium fractionation factor? What do you mean by cooling conditions?

- Conclusions
  - Line 459: "the deviation of hydrogen and oxygen isotope lines from the origin" As in the abstract, what are these lines and the origin?
  - Lines 468-471: Please review the cluster approach: I don't see how the clusters prove a linear relationship between $\delta D$ and $\delta^{18}O$.
  - Line 474: "the combined application of the IECEWV and Rayleigh fractionation models" Same comment as above, I don't understand what you mean by a combination of IECEWV and the Rayleigh model.

- References:
  - Please, check the doi-links in the references. Many links do not work due to a punctuation mark mistake.

**Minor comments**

- Line 78: quantitatively → do you mean "quantifying"?
- Line 85: deeply → do you mean "deepen"?
- Line 309: specifically → delete
- Line 349: stage → do you mean "plateau"?
- Line 366: in the article → I would refer to the section where you introduce the model.
- Lines 444-445:"In the future, water vapour isotopic values under saturated or near-saturated conditions should be calculated **without** in situ observations by combining this equation with observed raindrop spectral data." → Do you mean "with in situ observations"?

**References:**

Craig, H. and Gordon, L.: Deuterium and oxygen 18 variations in the ocean and the marine atmosphere, in: Proceedings of the Stable Isotopes in Oceanographic Studies and Paleotemperatures, 1965.

Dansgaard, W., 1954: The O18-Abundance in fresh water. Geochim. Cosmochim. Acta, 6, 241–260.

Stewart, M. K., 1975: Stable isotope fractionation due to evaporation and isotopic exchange of falling waterdrops : Applications to atmospheric processes and evaporation of lakes. J. Geophys. Res., 80, 1133–1146, doi:10.1029/JC080i009p01133.

---

## Author Comment (AC1)

Reviewers' comments:

Associate Editor: I have received 2 reviews back. The 2 reviewers are quite positive, even though they have a number of comments and suggestions that need to be addressed, the author believes that exchange model and the experimental results of this research are reasonable and credible. In order to enhance the reliability of the data quality, the authors added standard errors to the observations and calculations. In addition, the connection of Rayleigh model, equilibrium fractionation model and exchange model are also sorted out for further review by readers and reviewers.

The authors would like to thank all reviewers for their constructive comments and suggestions which have greatly improved this article. The point-by-point responses to all comments are given below. The parts in italic are the reviewer's comments, which are followed by our responses in blue.

Reviewer #1

*The manuscript "Theoretical and experimental investigation of isotopic exchange between water vapour and droplets under isothermal saturation conditions" introduces an approach to estimate the effect of isotopic exchange between water vapour and droplets in saturated conditions on the isotopic composition of water vapour. The theoretical approach allows to calculate the change in isotopic composition of water vapour during this process from environmental factors. The presented data show a similar trend in the isotopic composition from theory and experiment during a time period of 5min.*

*This study addresses an important question of the effect of isotopic exchange under saturated conditions between droplets and water vapour on the isotopic composition of water vapour. This effect can, for example, not be modeled by the Craig and Gordon (1965) model due to non-linear effects close to saturated conditions. The authors use an interesting experimental approach to test their model. But in its current state, the manuscript does not convincingly show the relevance of their results due to (i) missing definition of the analysed exchange process and how it differs from established terms such as isotopic equilibration between phases, (ii) missing reference to existing models such as equilibrium fractionation factors and Stewart model for the exchange between droplets and water vapour, and (iii) weak statistical analysis without quantitative uncertainty estimates of their experimental and modeled results. In the following, these three main comments are explained in more detail, together with further comments.*

*Main comments:*

*(1) What is the definition of exchange process? What is its role for equilibrium fractionation and what is, for example, the difference between equilibration between water vapour and the rain droplets (as e.g.mention on lines 76-77) and the exchange process?*

Response: Thank you for your comments and constructive suggestions. The exchange process refers to the process in which liquid droplets exchange with gaseous water molecules in a saturated condition (the same is true for the unsaturated and supersaturated state, but this study did not involve them). During this process, the molecular number of the liquid droplets and water vapour remain constant. The differences between equilibrium fractionation and the exchange process are mainly reflected in three aspects. First, equilibrium fractionation is a state and requires a process, and generally, an evaporation or condensation process occurs before the equilibrium fractionation is formed. Second, after the water vapour reaches saturation, due to the difference in molecular weight, different molecular types of water molecules ($H_2^{16}O$、$HD^{16}O$、$H_2^{18}O$) exhibit differential allocation between the gas phase and the liquid phase, before equilibrium is reached, molecular exchange occurs, which is the exchange process proposed in this study. Third, when the molecular allocation process in the liquid phase and gas phase reaches equilibrium, the exchange process reaches equilibrium, which is also the isotope equilibrium fractionation, in this state, although the isotope exchange process continues to exist, the phase distribution reaches equilibrium, so the exchange process does not cause changes in the isotope value. In summary, the exchange process is one way for isotope equilibrium fractionation to be achieved, and at present, the isotope equilibrium fractionation can only be calculated for the final state, not for the isotope changes in reaching the equilibrium fractionation process.

We have added the exchange process definition "the exchange process in the paper refers to the process in which liquid droplets exchange with gaseous water molecules in a saturated condition"(line 46), and also,

the difference between equilibrium fractionation and exchange model is added. The original text is "It should be noticed that the Rayleigh distillation model can calculate the isotope values for droplets and water vapour during the unsaturated conditions (relative humidity less than 100%), and the equilibrium fractionation model can only calculate the isotope values when the equilibrium fractionation is reached, but in reaching the equilibrium fractionation process-exchange process, the isotope changes cannot be calculated by the Rayleigh and equilibrium fractionation models."(lines 78-80)

*(2) The experimental results are only compared to the model introduced in this study. It is difficult to relate the results, especially the magnitude of the observed changes in δD and δ¹⁸O, to the fractionation effect as simulated by other, existing models (e.g. equilibrium fractionation effect, Craig-Gordon (1965) model, Stewart (1975) model). Some of these model cannot be applied at 100% relative humidity but any discussion of these limitations and how the theoretical approach of this study is filling a gap is missing. The study should outline more clearly how their results relate to previous theoretical models and frameworks.*

Response: Thank you for your suggestions. According to the above opinions, this paper first defines the exchange process, which is a process to go through in equilibrium fractionation. The differences between the models introduced in this study and those studied by predecessors are explained.

This study is different from the C-G model, which considers the evaporation and condensation processes and assumes that the ocean-atmosphere is a closed system (without considering land surface processes), and calculates the water isotope values of cloud layers based on the principle of water conservation. However, it pays little attention to the isotope changes caused by exchange process. Stewart's exchange model only calculates the isotope value of a single droplet and does not consider the isotope changes of water vapour exchanged with the droplet. It should be noted that equations (6) and (7) in this study are based on the principle of mass conservation and the Stewart model, the exchange equations between the liquid droplet and water vapour isotope are established in this study, and the equations are used to studying the time-varying isotope features of water vapour during the exchange process, which is an non-equilibrium exchange process and not in a state of equilibrium fractionation. In summary, this study fills the gap in the isotope features of the water vapour during exchange between the liquid droplets and water vapour (non-equilibrium exchange process).

We have added the limitations of existing models (e.g. equilibrium fractionation effect, Craig-Gordon (1965) model, Stewart (1975) model), and introduced our model. And the relationships of these models are also summarized, revised the original as follows:

It should be noticed that the Rayleigh distillation model can calculate the isotope values for droplets and water vapour during the unsaturated conditions (relative humidity less than 100%), and the equilibrium fractionation model can only calculate the isotope values when the equilibrium fractionation is reached, but in reaching the equilibrium fractionation process-exchange process, the isotope changes cannot be calculated by the Rayleigh and equilibrium fractionation models. Although the Stewart (1975) model gives the isotope changes of the exchanging droplet, the single droplet exchange scenario is not consistent with

the actual droplets and water vapour exchange scenario, and the water vapour isotope changes during the exchange process have not been detected. Therefore, this study references the exchange model (Bolin, 1958; Friedman et al., 1962; Miyake et al., 1968; Stewart, 1975), combines the mass conservation during the exchange process, gives the water vapour isotope calculation equation during the exchanging between droplets and water vapour.(lines 76-85)

We use the equilibrium fractionation model to calculate the fractionation coefficient after reaching equilibrium exchange, in order to verify the isotope calculation model introduced in this study.

According to the equilibrium fractionation model, at the temperature of 20℃, the $^{18}$O and $^2$H equilibrium fractionation coefficients calculated by Majioube (1971) model are 1.0098 and 1.085, respectively, and according to the equilibrium fractionation calculation formula, there are

$$\alpha_{l\text{-}v} = \frac{1+\delta_l}{1+\delta_v}$$                                                (1)

Where $\alpha_{l\text{-}v}$ is the equilibrium fractionation coefficient, and $\delta_l$ and $\delta_v$ are the isotopic values of liquid and gaseous water under equilibrium fractionation conditions, respectively. Since the water vapour isotope value was not mentioned in Stewart model, the isotopic fractionation coefficient calculated by our model was compared with Majioube (1971) and Stewart (1975) models, revised the original as follows:

In order to verify the reliability of this model, the fractionation coefficient is calculated and compared with the existing models in the exchange equilibrium stage (the isotope equilibrium fractionation stage). The $^2$H and $^{18}$O equilibrium fractionation coefficients given by Stewart (1975) model based on exchange and evaporation were 1.114±0.005 and 1.0288±0.001, respectively. According to the observation results of this experiment, the isotopic values of liquid water are taken as the average isotope values, and their average values are -39.74‰ and -6.85‰ for $\delta^2$H and $\delta^{18}$O, respectively, and the average values of water vapour are -84.01‰ and -13.36‰ for $\delta^2$H and $\delta^{18}$O (equilibrium exchange state, 219-300 s), respectively. Therefore the values of $^2$H and $^{18}$O equilibrium fractionation coefficients in this experiment are 1.048±0.0056 and 1.0065±0.001, respectively. The values of $^2$H and $^{18}$O equilibrium fractionation coefficients given by Majioube (1971) are 1.085 and 1.0098, respectively. The relatively errors of $^2$H and $^{18}$O equilibrium fractionation coefficients are 3.41% and 0.32% (Majioube (1971) value as standard), respectively.(lines 410-419).

 (3) Statistical analysis: The discussion of the difference between the experiment and the modeled data focuses mainly on the relative difference between the two datasets. This discussion does not address the uncertainty of the experiment. Do the differences fall within the uncertainty and are the changes during the 300s significant with respect to the differences between different model runs? Further, the cluster analysis

*is not documented and the interpretation of the clusters as a linear relationship between δD and $\delta^{18}O$ is questionable .*

Response: Thank you for your comments. We have addressed all the issues raised in the comments. The data quality control is of great importance for the reliability of the research conclusions. The paper discusses the uncertainty of the experiment and theory in this study, but does not provide specific quantification. The author has supplemented the standard error of the experimental observation means in the paper (the author has also modified the figure in the paper). As shown in the figure 1, it can be found that the trend of the upper and lower limits of the standard error of the experimental observation values is consistent with experimental observations, so the fluctuation of the experimental observation means are not caused by random errors in the experimental observations, the trend of the experimental observation mean has representativeness. However, the fluctuation of the experimental observations is within the range of experimental standard error (SE), which may be related to the fact that the number of water vapour isotope observation points is too small (under the existing observation conditions, we can only continuously observe the water vapour isotope value of one point) (This part is added after the result description in Figure 4 of manuscript, on lines 333-338). The author will design a more comprehensive experiment to further verify it in the future.

[Figure]

Figure 1 Variation of hydrogen and oxygen isotopes over time

We also present the graph of standard error (SE) over time, as shown in Figure 2, where it is clearly evident that the SE of $\delta^2H$ observations fluctuates with a pattern of first decreasing and then stabilizing over time. During 0-109s, the SE data is larger, indicating that the isotope exchange process fluctuates greatly during the initial period of sound wave action to a droplet sedimentation cycle, which is attributed to the increase in particle size and adjustment of isotope distribution. During 109 s-218 s, the exchange process in the exchange chamber is relatively stable, and the SE decreases. Between 218s-300s, the exchange process reaches stability, and the observed SE values are smaller. The SE of $\delta^{18}O$ observations also fluctuates over time, with a mean value that remains essentially unchanged.

[Figure]

Figure 2. Variation of standard error of observation over time

Regarding the discussion of isotope linear relationships, the author found that theoretical and experimental observations show that the $d$-excess remains constant during isotope exchanging, but meteoric water line (instead of linear relationship between $\delta$D and $\delta^{18}$O) after exchanged cannot be derived from theoretical formulas due to the influence of the exchanged droplets and the initial isotope values of water vapour, but the meteoric water line is of greater concern in the water cycle process, so the author combined the observed values and theoretical calculated values to fit the observed values. Based on your advice, the paper finally provides the theoretical water vapour isotope relationship during the exchange process, as well as the actual distribution of water vapour isotopes observed, and downplays the discussion and description of the meteoric water line. Revised original text as follow:

A linear relationship fit to the calculated hydrogen and oxygen isotope values, the MWL was $\delta^2$H=7.19$\delta^{18}$O+12.14 ($n$=95, $R^2$=1.0, $p$=0.05). The slope was slightly smaller than that of the GMWL, but the intercept was slightly larger. A linear cluster with the same slope was formed (Figure 8(b)), and the experimental points were well distributed around the linear cluster (four lines with a slope of 7.19, and intercepts for the horizontal axis from left to right were -13.392, -13.365, -13.338, and-13.305, with a bandwidth of 0.027)(line 453-457).

*(4) The manuscript is often difficult to read due to incomplete sentences, incorrectly used words and grammatical inconsistencies. I do not list all of these issues below, please thoroughly check the language of the manuscript again.*

Response: Thanks, we have carefully checked the manuscript for language and grammatical errors, and asked the professional language company for further correction.

*(5) The figure captions are often too short. They lack information on the shown data or schematics. Please, improve the captions such that the reader can understand the figures by reading the captions without reading the text.*

Response: Thanks. All the figure captions have been checked and added the relevant descriptions to improve the readability.

*(6) There are several terms used in the abstract that are not defined/generally known in the field: exchange process (line 17), hydrogen and oxygen isotope line (line 25), the origin (line 25).*

Response: The definition of the exchange process has been added on lines 17-18 (refers to the process in which liquid droplets exchange with gaseous water molecules in a saturated condition), hydrogen and oxygen isotope line has been modified to meteoric water line (MWL) (line 29), the origin has been modified to the coordinate origin (line 29).

*(7)It is difficult to follow the description of the results in the abstract without having read the manuscript.?*

Response: I an sorry for my abstract is difficulty to understand without having read manuscript. We have revised the abstract. After introducing the isotope exchange process studied in this paper. In the part of experimental results, the three stages of water vapour isotopes change and the causes of each corresponding stage are described in detail. Then, the variation trend and relative error of the calculated results of our model are explained. Finally, our model is used to analyze the change of *d*-excess during the exchange process and its effect on the meteoric water line. Revised original text as follow:

Abstract: Water cycle process will deeply affect water vapour isotope composition, in addition to the evaporation and condensation processes, the exchange process (refers to the process in which liquid droplets exchange with gaseous water molecules in a saturated condition) is also a crucial process that influences isotopic variations. To explore the mechanism of isotope exchange between droplets and water vapour under isothermal saturation conditions, we developed an isotope exchange calculation equation for water vapour (IECEWV) based on the conservation of exchange quantity, we use acoustic waves to increase the droplets sizes and conduct indoor trials with 100 control groups to validate the IECEWV. The results indicated that at the initial of the droplet sizes increased, the degree of isotope distribution in droplets increased which leading to the water vapour isotopic enrichment, and then the specific surface area of the droplets become the control factor of the exchange process and thus caused the isotopes depletion, finally, the isotope values are basically stable resulted in equilibrium exchange. For the exchange process controlled by specific surface area, the IECEWV calculated isotopes capture the observed values, with average maximum mean relative errors of 1.66 % for $\delta^2$H and 3.19 % for $\delta^{18}$O, and the analysis of frequency distribution for 10,355 calculated relative errors shows the relative errors were primarily concentrated within 4.5 %, with a cumulative frequency exceeding 98 % for relative errors less than 9.5 %. IECEWV analysis indicates isotopes exchange can lead to the meteoric water line (MWL) deviate from the coordinate origin, furthermore, the d-excess of water vapour remains essentially constant. Future studies may combine the IECEWV with the Rayleigh fractionation model to explore the coordinated changes in precipitation and atmospheric water vapour isotopes.(lines 16-31)

*(8) Line 62: " in an isotope exchange scenario" do you mean "in an isotope exchange-only scenario*

Response: Yes, the authors want to express the effect of isotope exchange on the MWL of water vapour, but simple actual exchange is rare in nature. This sentence has been revised to "It is worth noting that the MWL is an isotopic relationship involving non-equilibrium fractionation, equilibrium fractionation and exchange process, but the impact of isotope exchange-only scenario on MWL and d-excess is still unclear. "(lines 64-66)

*(9) Line 67: "Gu et al. 2011" Can you add a reference to earlier work on this topic?.*

Response: By reviewing the literature, the author cited the literature with high citation rate and revised it in the paper. This sentence has been revised to "Owing to their disparate molecular activities, heavier molecules tend to exhibit relatively enriched concentrations in the liquid phase and depleted concentrations in the gaseous phase (Mook, 2001)" (lines 68-69)

Mook, W.G.: Environmental isotopes in the hydrological cycle: principles and applications. In: V. I: Introduction, Theory, Methods, Review, IAEA-UNESCO, 1–165 pp, 2001.

*(10) Lines 70-71: "kinetic and diffusion fractionation" What is the difference between kinetic and difussion fractionation?*

Response: Sorry for that, our expression is not precise enough. Dynamic fractionation mainly refers to the process of deviation from equilibrium fractionation, which is closely related to time, such as the change of reaction system temperature, dynamic fractionation will occur. Isotope diffusion fractionation mainly refers to the isotope fractionation caused by isotopic concentration gradient. Diffusion fractionation occupies a major position at the beginning of the reaction of the system (the initial stage of the evaporation and condensation process), and dynamic fractionation occupies a major position at the later stage of the evaporation and condensation process (changes in pressure and temperature with time). To some extent, there is little difference between dynamic fractionation and diffusion fractionation, but both belong to non-equilibrium fractionation. In order to avoid too much conceptual confusion, the authors unify the non-equilibrium fractionation in the paper.

*(11) Lines 71-72: "In many cases, the Rayleigh fractionation model is utilised to describe such nonequilibrium processes (Liu et al., 2015)", The Rayleigh model was introduced by Dansgaard (1954).*

Response:Thanks for pointing out the error! You're absolutely right. We've corrected the citations. You can find it on lines 71.

*(12) The Rayleigh model describes the evolution of the isotopic composition of water vapour and rain during a rain-out process. This can involve non-equilibrium fractionation if the cloud droplets are forming in over-saturated conditions. But I would not use it as typical example of non-equilibrium fractionation?*

Response: Indeed, if the precipitation formed in the over-saturated conditions, this process includes equilibrium fractionation and non-equilibrium fractionation, the author has revised this sentence to "However, non-equilibrium fractionation occurs predominantly in actual processes such as condensation process under unsaturated conditions (Gu et al., 2011)"(lines 72-73).

*(13) Lines 76-77: " the isotope exchange process leads to weaker equilibration on the water vapour and precipitation isotope values (Graf et al., 2019)" what do you mean by weaker equilibration?*

Response: The author wants to express that isotope exchange lead to isotope equilibration (there is an effect on the isotope value, but the data change is relatively small), and the original expression has been modified to "the isotope exchange process leads to equilibration on the water vapour and precipitation isotope values (Graf et al., 2019)"(lines 55-56).

*(14) Lines 85-87:"The study will deeply our understanding of isotopic changes in the water cycle, especially for coordinated changes in precipitation and water vapour isotopes, and further expand the*

*applicable accuracy and scope of the Rayleigh fractionation model." Why are results of this study (especially) relevant for the Rayleigh fractionation model?*

Response: Thanks. Rayleigh fractionation model is a non-equilibrium fractionation, but in the evolution process from non-equilibrium fractionation to equilibrium fractionation, the final equilibrium fractionation will be formed through isotope exchange. In the actual precipitation process in the field, there must be isotope exchange process before equilibrium fractionation is reached, so the combination of isotope exchange process and Rayleigh fractionation model in this paper will play an important role in further refining the investigation of isotope changes in the precipitation process.

*(15) The discussion of literature stays mostly on a macroscopic level without further discussing the processes on a molecular level even though these processes are important in the methods and results section. How has water vapour – droplet interaction been discussed in literature? How does the approach of this study differ from previous approaches?*

Response: At present, it is generally believed that the isotope exchange process occurs in moleculars (Thibblin et al,1989; Pyper et al., 1967; Wahl et al., 2021) between identical phases, and for the exchange between different phase states (liquid and vapour), we assume that the exchange is carried out by the water molecule as the smallest unit, because the reaction in the process of evaporation and condensation is also carried out by the molecule as the smallest unit (We also add the above discussion to the assumptions which can be found on lines 116-119). When the exchange at the molecular level reaches equilibrium, the macro isotope balance is also reached. The difference between the paper and the previous research is that acoustic wave technology is used to promote the collision of droplets, resulting in the increase of particle size, and the isotope exchange process between droplets and the water vapour during the growth of particle size is simulated. Second, in situ monitoring was used to give the characteristics of water vapour isotope changes in real time, while previous studies focused more on droplet isotope changes.

A. Thibblin, P. Ahlberg, Reaction branching and extreme kinetic isotope effects in the study of reaction mechanisms. Chem. Soc. Rev. 18, 209–224,1989.

J. W. Pyper, R. S. Newbury, G. W. Barton Jr., Study of the isotopic disproportionation reaction between light and heavy water using a pulsed-molecular-beam mass spectrometer. J. Chem. Phys. 46, 2253–2257 ,1967.

Wahl, S., Steen-Larsen, H. C., Reuder,J., & Hörhold, M. Quantifying the stable water isotopologue exchange between snow surface and lower atmosphere by direct flux measurements. Journal of Geophysical Research: Atmospheres,126, e2020JD034400. https://doi.org/10.1029/2020JD034400,2021.

• *Methods and materials*

*(16) In this section, past and present tenses mix and it is not always clear why the tense changes. Please, check again their usage.*

Response: In the case of constant temperature and relative humidity, in order to describe the isotope exchange process, this study also simulated the characteristics of particle size change, which is related to the time of sound wave, so the time change process is described. As for the tense put together, the expression is not clear, the author has modified the expression and the picture.

*(17)Fig.1: The schematic of the experiment (green box) is difficult to understand. What are the numbers? What are Type? What do the arrows mean?*

Response: Thanks. The experiment in Figure 1 in manuscript mainly expresses the in-situ observation of the critical parameter of the experiment, and the mumbers are intended to express the specific observation device (Figure 2 in manuscript, to be associated with the diagram of the experimental device). However, it may be difficult to read, and the author has also modified the picture( as follow), please review it again.

[Figure]

Figure 3: Correlations between theory and experiment of the article. Under the isothermal saturation conditions, the droplets will exchange molecules with water vapour, and the total number of molecules in the droplets and water vapour will remain unchanged during the exchange process. According to the conservation of mass (conservation of molecular number), the isotopic relationship between droplets and water vapour was established, and the change of water vapour isotope in the exchange process was expressed. Through laboratory experiments, the changes of liquid droplets and water vapour isotopes after exchange were monitored, and the theoretical water vapour isotope values were verified. (N stands for number of molecules, subscripts D and V stand for droplets and vapour, subscript H stands for heavy molecules such as HDO or $H_2^{18}O$, t is time, C is constant, T is temperature, R stands for isotope ratio, RH relative humidity)

*(18) Equation (1): more explanation is needed how equation 1 is derived. What does it represent? Does follow from the assumption that the rate of change in water molecules in the vapour phase and droplets are equal?*

Response: Thanks, formula (1) expresses the mass conservation equation in the exchange process, that is, the rare molecular number increased in the droplet is equal to the rare molecular number decreased in the water vapour, that is, the exchange number conserved in the exchange process, and the explanation is added in the paper to better understand the physical meaning of the equation (1), which are "The left and right sides of the equation represent the increased number of rare isotope molecules in the droplets and decreased number of rare isotope molecules in water vapour, respectively, $N_D R_{D0}/(1+ R_{D0})$ represents the total number of rare isotope molecules in the droplet at the initial moment, and the rest is similar."(lines 136-138).We do not assume that the molecular number of water vapour and the droplet change at the same rate, the subsequent isotopic ratio change is derived by using the equation.

*(19) Equation (4): What is Rr?*

Response: $R_r$ is ratios of the standard mean ocean water, which is a constant, and for $^2H$ and $^{18}O$, the values are $(2005.2 \pm 0.45) \times 10^{-6}$ and $(155.6 \pm 0.12) \times 10^{-6}$, respectively, which have been added in the paper.

*(20) Equation (6): What is γ?*

Response: $\gamma$ is a coefficient that is related to the droplet size (added on line 157), the value is 1.0 in this paper.

*(21)Line 166: "significant reduction": What is a significant reduction? Did you do a statistical test?*

Response: The author wants to express the obvious decline, it is a theoretical analysis of the isotope exchange equation, and does not do a significance test. It has been modified to obvious reduction.

*(22) Line 166-167: "a faster reduction in the specific surface area of the droplets indicated a higher rate of droplet coalescence, which resulted in a significant decrease in water vapour isotopes."Is this still theoretical or does this refer to the experiment?*

Response: The authors want to express the rapid decline, it is a theoretical analysis of the isotope exchange equation, and we do not do a significance test. The sentence has been revised to "a faster reduction in the specific surface area of the droplets indicated a higher rate of droplet coalescence, which resulted in a rapid decrease in water vapour isotopes."

*(23) Line 192: "it remains essentially constant" What is "it"?*

Response: It refers to the isotopic gradient, including $(\delta^{18}O_D-\delta^{18}O_V)$ and $(\delta^2H_D - \delta^2H_V)$, which has been revised and can be found on line 206.

*(24)Fig.2 caption: " I includes 1 and 3, is exchange chamber 4, includes 9-13, includes 2, 5-8  II  III  IV and 14-15)" What are I, II, III and IV?*

Response: The experimental setup consisted of a droplet generation system (I), exchange chamber (II), online particle size monitoring, temperature, and humidity measurement (III), and isotopic monitoring device (sampling and test) (IV), which has been added in figure caption.

*(25) Fig. 2: The inlet of the water vapour analysis (5) is above the droplet inlet (3). How do you make sure that the droplets distribute through the entire chamber before they settle?*

Response: Thanks. The droplets forms a conical surface from the outlet, the apical angle is 30°, the observed droplet velocity of 2 m/s (Shi et al, 2021), and the outlet is 1 m away from the top of the exchange chamber,

the outermost spreading distance of the droplet is 54 cm, and the diameter of the exchange chamber is 60 cm, so the droplet can fill the entire exchange chamber.

Shi, Y., Wei, J., Bai, W., and Wang, G.: Numerical investigations of acoustic agglomeration of liquid droplet using a coupled CFD-DEM model, Adv. Powder Technol., 31, 2394-2411, https://doi,org/10.1016/j.apt.2020.04.003, 2020.

*(26) Lines 273-274: "Over 60,000 data points for isotopes and droplet particle sizes were produced during the procedure, including 400 liquid water samples." Why do you mention these numbers? They do not help to understand the outcome of the experiments. It would, e.g. be more useful to know the number of experiment runs.*

Response: This data is provided to show that the research conclusion of this paper is supported by a large number of reliable data. 100 groups of repeated experiments were completed in this experiment, we also add the number of the experiment runs in paper and can be found on line 261.

*(27)Lines 278-279 and lines 283-284: "The characteristic particle size was averaged over experimental runs" and "The water vapour isotope data were averaged using the same approach as that used for the aforementioned characteristic particle size." How was this averaging done? Are all runs averaged to derived mean temporal evolutions of the measured quantities during 300s runs?*

Response: The experiment was carried out in a cycle, that is, the sound waves intervened - the sound waves did not intervene, and so on. In the whole experiment, 100 groups of control experiments were carried out, and 95 groups were used for data analysis, and 5 groups of incomplete and significant variation data were excluded. The average data of the 300s of sound wave intervention is obtained in this way, that is, the data of the first 1s is the average data of the 95 groups after sound wave interfering for 1s, the data of the second 2s is the average data of the 95 groups after sound wave interfering for 2s, and so on. And we also revised the droplet size date processing as follow:

The experimental mean value of acoustic wave intervention and natural conditions can be calculated as follows: First, the time average of each group (300 s) is calculated, and then the average value of all experimental groups was calculated to obtain the average particle size spectrum data. Additionally, to elucidate the process of particle size variation with temporal, the characteristic particle size of all acoustic intervention groups were averaged over experimental runs, resulting in temporal data for the variation in $D_{90}$ (This means that the volume frequency of droplets smaller than this particle size accounts for 90 % of the total volume) over time, that is, the data at 1s is the average data of all groups after sound wave interfering for 1s, the data at 2s is the average data of all groups after sound wave interfering for 2s, and so on. Similarly, the mean value of specific surface area was obtained over time.(lines 286-293)

• *Experiment results and analysis*

*(28) Line 299: "increase/decrease" What does this mean? Please, be more specific.*

Response: Referring to the characteristic particle size and specific surface area respectively, this sentence has been modified to "the characteristic droplet size ($D_{90}$) with an initial increase followed by stabilisation, but specific surface area ($S_{AT}$) with an initial decrease followed by stabilisation."(lines 310-311).

(29) *Lines 303-305: This sentence is too long.*

Response: Thanks. This sentence has been modified, which is "A larger specific surface area indicates a greater potential for isotopic exchange between the droplets and water vapour, when the saturated exchange state was not reached"(lines 315-317).

(30)*Fig. 3-5: Do you show means over all experimental runs in Fig. 3-5 means? Can you add a standard deviation (or some uncertainty estimate) around the mean value?*

Response: Thanks to the reviewer, the data in Figure 3-5 are average values, and standard error has been added to the original figures.

*(31) Fig. 3: What are the vertical dashed lines in (a)?*

Response: The dashed lines are to elicit and express the characteristic particle size $D_{90}$, $D_{90}$ values in the acoustic intervention and natural conditions are 63 μm and 45 μm, respectively. We added an explanation of dashed lines in the Figure 3 captions and can be found on lines 320-322.

*(32)Fig. 3: The changes in δD and δ¹⁸O are relatively small in this experiment. Is this caused by the small difference in the isotopic composition between droplets and vapour? What magnitude of changes do you expect if the isotopic composition differs more strongly between the droplet and its surrounding?*

Response: Thanks to the expert's keen and meticulous observation, the observed isotopic variation in this paper is relatively small, but the difference between the isotopic values of droplets and water vapour is still relatively large. Because of the difference in phase state, the isotopic values are different by an order of magnitude. The author also lists the isotopic data values of liquid water and water vapour for calculating the isotopic gradients (table 1). If the isotopic composition differs more strongly between the droplet and its surrounding, the variation of water vapour isotope values (before and after exchange) may be outside the error range, but the variation trend is the same as that in this experiment.

Table 1 liquid water and water vapour isotope values

| Experimental number | Vapour $\delta^{18}O$(‰) | Droplets $\delta^{18}O$(‰) | Droplets $\delta^{18}O$ SD | Vapour $\delta^{2}H$(‰) | Droplets $\delta^{2}H$(‰) | Droplets $\delta^{2}H$ SD | $\delta^{18}O_D - \delta^{18}O_V$ (‰) | $\delta^{2}H_D - \delta^{2}H_V$ (‰) |
|---|---|---|---|---|---|---|---|---|
| 1 | -16.14 | -7.39 | 0.33 | -98.64 | -43.70 | 1.53 | 8.75 | 54.94 |
| 2 | -16.02 | -7.51 | 0.23 | -95.51 | -43.95 | 0.95 | 8.51 | 51.56 |
| 3 | -14.99 | -7.51 | 0.07 | -90.24 | -43.14 | 0.48 | 7.49 | 47.10 |
| 4 | -15.98 | -7.78 | 0.38 | -91.34 | -44.37 | 2.61 | 8.20 | 46.96 |
| 5 | -14.63 | -7.63 | 0.33 | -88.64 | -44.47 | 1.52 | 7.00 | 44.18 |
| 6 | -14.38 | -7.33 | 0.27 | -88.01 | -43.34 | 1.39 | 7.05 | 44.67 |
| 7 | -14.49 | -7.78 | 0.27 | -85.74 | -44.19 | 1.57 | 6.71 | 41.56 |
| 8 | -13.84 | -7.51 | 0.11 | -80.56 | -43.75 | 0.59 | 6.32 | 36.81 |
| 9 | -13.51 | -7.41 | 0.09 | -83.49 | -43.57 | 0.79 | 6.11 | 39.92 |
| 10 | -13.39 | -7.40 | 0.10 | -82.10 | -43.54 | 0.15 | 5.99 | 38.56 |
| 11 | -13.88 | -7.40 | 0.36 | -84.72 | -42.76 | 2.12 | 6.48 | 41.96 |

| | | | | | | | |
|---|---|---|---|---|---|---|---|
| 12 | -14.57 | -7.33 | 0.13 | -82.27 | -42.63 | 1.02 | 7.24 | 39.64 |
| 13 | -13.64 | -7.63 | 0.12 | -82.74 | -44.34 | 1.05 | 6.00 | 38.40 |
| 14 | -12.59 | -7.61 | 0.07 | -79.75 | -43.61 | 0.71 | 4.99 | 36.14 |
| 15 | -14.19 | -7.56 | 0.09 | -87.31 | -43.52 | 1.21 | 6.63 | 43.79 |
| 16 | -12.87 | -7.21 | 0.42 | -79.14 | -42.38 | 2.57 | 5.66 | 36.76 |
| 17 | -13.33 | -7.44 | 0.12 | -83.74 | -42.30 | 1.08 | 5.90 | 41.44 |
| 18 | -13.94 | -7.54 | 0.29 | -82.47 | -41.82 | 1.75 | 6.40 | 40.65 |
| 19 | -13.44 | -7.33 | 0.06 | -82.67 | -43.75 | 0.26 | 6.11 | 38.93 |
| 20 | -13.57 | -7.55 | 0.17 | -82.33 | -43.63 | 0.65 | 6.02 | 38.70 |
| 21 | -13.95 | -7.99 | 0.26 | -83.19 | -43.72 | 1.33 | 5.96 | 39.47 |
| 22 | -14.65 | -7.60 | 0.33 | -85.05 | -43.79 | 2.38 | 7.05 | 41.26 |
| 23 | -13.39 | -7.11 | 0.11 | -83.92 | -42.56 | 0.70 | 6.28 | 41.36 |
| 24 | -12.93 | -7.17 | 0.04 | -82.17 | -41.90 | 0.33 | 5.76 | 40.27 |
| 25 | -13.37 | -7.21 | 0.07 | -80.15 | -42.91 | 0.24 | 6.17 | 37.24 |
| 26 | -14.36 | -7.53 | 0.14 | -84.39 | -44.46 | 1.05 | 6.84 | 39.94 |
| 27 | -14.38 | -7.45 | 0.50 | -80.96 | -43.62 | 3.83 | 6.93 | 37.34 |
| 28 | -13.64 | -7.67 | 0.79 | -82.30 | -43.92 | 5.44 | 5.97 | 38.38 |
| 29 | -14.58 | -7.31 | 0.41 | -86.92 | -42.35 | 3.21 | 7.26 | 44.57 |
| 30 | -14.17 | -7.32 | 0.13 | -83.99 | -43.12 | 1.01 | 6.85 | 40.87 |
| 31 | -13.51 | -7.27 | 0.79 | -79.02 | -43.36 | 4.95 | 6.24 | 35.66 |
| 32 | -13.24 | -7.67 | 0.40 | -82.38 | -44.67 | 3.72 | 5.57 | 37.72 |
| 33 | -14.08 | -7.38 | 0.31 | -84.00 | -43.89 | 3.78 | 6.70 | 40.12 |
| 34 | -15.10 | -7.12 | 0.13 | -86.23 | -43.10 | 1.10 | 7.98 | 43.12 |
| 35 | -13.73 | -7.43 | 0.41 | -84.19 | -43.05 | 2.79 | 6.31 | 41.14 |
| 36 | -14.31 | -7.40 | 0.13 | -82.67 | -42.94 | 0.96 | 6.91 | 39.73 |
| 37 | -14.31 | -7.41 | 0.14 | -84.84 | -43.35 | 1.09 | 6.90 | 41.49 |
| 38 | -14.25 | -6.92 | 0.24 | -83.86 | -40.68 | 1.92 | 7.33 | 43.18 |
| 39 | -12.81 | -7.09 | 0.44 | -80.31 | -42.05 | 2.25 | 5.72 | 38.26 |
| 40 | -14.16 | -7.05 | 0.35 | -84.83 | -40.96 | 3.70 | 7.11 | 43.86 |
| 41 | -13.79 | -7.35 | 0.31 | -83.17 | -42.77 | 2.97 | 6.43 | 40.39 |
| 42 | -13.97 | -7.32 | 0.09 | -82.90 | -43.44 | 0.92 | 6.64 | 39.46 |
| 43 | -14.19 | -7.32 | 0.12 | -82.70 | -43.23 | 0.62 | 6.87 | 39.47 |
| 44 | -13.93 | -6.73 | 0.35 | -85.01 | -38.37 | 1.75 | 7.20 | 46.64 |
| 45 | -13.11 | -6.85 | 0.52 | -83.86 | -39.11 | 3.43 | 6.26 | 44.75 |
| 46 | -13.97 | -7.11 | 0.32 | -86.28 | -41.69 | 1.49 | 6.86 | 44.59 |
| 47 | -13.15 | -6.55 | 0.27 | -82.65 | -38.75 | 2.82 | 6.61 | 43.90 |
| 48 | -14.60 | -7.35 | 0.74 | -87.34 | -42.52 | 5.04 | 7.25 | 44.83 |
| 49 | -12.09 | -7.44 | 0.36 | -75.14 | -43.67 | 2.86 | 4.65 | 31.47 |
| 50 | -13.91 | -7.45 | 0.23 | -81.16 | -43.40 | 2.37 | 6.47 | 37.76 |
| 51 | -13.91 | -7.34 | 0.07 | -84.97 | -43.43 | 0.31 | 6.57 | 41.54 |
| 52 | -13.29 | -7.36 | 0.08 | -81.40 | -43.29 | 0.40 | 5.93 | 38.10 |
| 53 | -13.50 | -7.36 | 0.13 | -83.12 | -42.79 | 0.64 | 6.14 | 40.33 |
| 54 | -13.47 | -7.48 | 0.23 | -86.30 | -43.58 | 1.01 | 5.99 | 42.72 |
| 55 | -13.18 | -7.27 | 0.05 | -83.99 | -43.09 | 0.33 | 5.91 | 40.91 |

| | | | | | | | |
|---|---|---|---|---|---|---|---|
| 56 | -14.11 | -7.25 | 0.13 | -85.73 | -43.05 | 0.77 | 6.86 | 42.67 |
| 57 | -13.67 | -7.70 | 0.03 | -83.58 | -43.86 | 0.33 | 5.97 | 39.72 |
| 58 | -12.73 | -7.38 | 0.14 | -98.87 | -43.01 | 1.21 | 5.35 | 55.86 |
| 59 | -12.35 | -7.30 | 0.12 | -95.49 | -43.32 | 1.10 | 5.06 | 52.18 |
| 60 | -11.89 | -7.54 | 0.09 | -94.91 | -42.24 | 0.51 | 4.35 | 52.68 |
| 61 | -11.83 | -7.36 | 0.03 | -92.12 | -42.46 | 0.72 | 4.47 | 49.67 |
| 62 | -11.83 | -7.43 | 0.12 | -91.50 | -43.06 | 1.05 | 4.40 | 48.45 |
| 63 | -11.99 | -7.50 | 0.58 | -91.06 | -43.07 | 3.56 | 4.48 | 47.99 |
| 64 | -11.70 | -7.46 | 0.00 | -91.08 | -42.88 | 0.00 | 4.24 | 48.19 |
| 65 | -10.89 | -7.29 | 0.58 | -86.77 | -41.29 | 4.33 | 3.59 | 45.48 |
| 66 | -13.67 | -7.33 | 0.00 | -95.58 | -40.57 | 0.00 | 6.34 | 55.00 |
| 67 | -12.96 | -7.46 | 0.11 | -91.63 | -42.43 | 1.10 | 5.50 | 49.20 |
| 68 | -12.81 | -7.44 | 0.29 | -90.04 | -42.79 | 2.33 | 5.37 | 47.25 |
| 69 | -12.25 | -7.39 | 0.12 | -80.70 | -42.16 | 0.64 | 4.86 | 38.54 |
| 70 | -12.58 | -7.46 | 0.61 | -88.95 | -42.32 | 4.22 | 5.11 | 46.63 |
| 71 | -12.60 | -7.51 | 0.41 | -87.16 | -41.42 | 2.99 | 5.09 | 45.73 |
| 72 | -12.88 | -6.98 | 0.49 | -87.65 | -40.88 | 4.01 | 5.90 | 46.78 |
| 73 | -12.61 | -7.42 | 0.10 | -80.62 | -42.76 | 0.59 | 5.19 | 37.86 |
| 74 | -12.84 | -7.49 | 0.12 | -82.59 | -42.51 | 0.94 | 5.35 | 40.08 |
| 75 | -12.99 | -7.38 | 0.68 | -80.71 | -43.48 | 4.98 | 5.61 | 37.23 |
| 76 | -13.25 | -6.96 | 0.35 | -86.91 | -43.52 | 2.78 | 6.29 | 43.40 |
| 77 | -12.73 | -6.72 | 0.21 | -76.90 | -43.29 | 0.97 | 6.01 | 33.61 |
| 78 | -12.84 | -6.91 | 0.13 | -80.87 | -43.81 | 0.75 | 5.94 | 37.06 |
| 79 | -12.35 | -6.94 | 0.20 | -78.52 | -44.25 | 1.83 | 5.40 | 34.27 |
| 80 | -13.50 | -7.06 | 0.37 | -82.07 | -43.99 | 2.55 | 6.44 | 38.08 |
| 81 | -12.73 | -7.15 | 0.12 | -79.34 | -44.01 | 0.83 | 5.58 | 35.34 |
| 82 | -12.66 | -7.39 | 0.03 | -78.83 | -44.28 | 0.39 | 5.27 | 34.55 |
| 83 | -13.97 | -7.58 | 0.06 | -87.01 | -44.10 | 0.02 | 6.39 | 42.91 |
| 84 | -12.22 | -8.30 | 0.00 | -76.56 | -47.23 | 0.00 | 3.93 | 29.33 |
| 85 | -13.22 | -7.97 | 0.00 | -86.47 | -45.19 | 0.00 | 5.25 | 41.28 |
| 86 | -13.27 | -7.82 | 0.00 | -78.95 | -43.68 | 0.00 | 5.45 | 35.28 |
| 87 | -12.05 | -7.23 | 0.23 | -78.94 | -40.70 | 2.09 | 4.82 | 38.24 |
| 88 | -12.30 | -7.72 | 0.48 | -76.59 | -43.02 | 3.07 | 4.58 | 33.56 |
| 89 | -12.66 | -7.66 | 0.00 | -79.51 | -43.66 | 0.00 | 5.00 | 35.84 |
| 90 | -13.39 | -7.63 | 0.17 | -84.06 | -43.14 | 1.13 | 5.76 | 40.92 |
| 91 | -12.64 | -7.99 | 0.61 | -78.77 | -44.53 | 3.62 | 4.65 | 34.24 |
| 92 | -13.40 | -7.71 | 0.07 | -79.86 | -44.10 | 1.16 | 5.69 | 35.75 |
| 93 | -12.73 | -7.93 | 0.18 | -79.47 | -45.17 | 1.41 | 4.80 | 34.29 |
| 94 | -12.86 | -7.61 | 0.13 | -83.25 | -44.66 | 1.19 | 5.25 | 38.59 |
| 95 | -13.02 | -7.30 | 0.14 | -83.83 | -44.27 | 0.49 | 5.73 | 39.55 |
| 96 | -12.27 | -7.03 | 0.13 | -77.81 | -44.36 | 0.59 | 5.24 | 33.45 |
| 97 | -13.03 | -6.96 | 0.55 | -81.17 | -41.25 | 3.25 | 6.07 | 39.92 |
| 98 | -12.67 | -6.88 | 0.37 | -82.28 | -41.45 | 1.80 | 5.78 | 40.83 |
| 99 | -12.52 | -6.93 | 0.10 | -79.18 | -43.38 | 1.33 | 5.58 | 35.80 |

| 100 | -11.27 | -7.10 | 0.19 | -76.88 | -41.51 | 1.58 | 4.17 | 35.37 |
|-----|--------|-------|------|--------|--------|------|------|-------|

*(33)Lines 342-343: "As the droplet size increases, the degree of rare isotope distribution on the surface of the liquid droplets increases, disrupting the previously established equilibrium exchange state." Why does the distribution of rare isotopes change with an increase in droplet size? What is the assumption on the isotopic distribution in a droplet? Is it well-mixed? Or is there a gradient between surface and inner part of the droplet?*

Response: Thanks for your valuable comments. The degree of rare molecule distribution on the surface of the droplet is the molecular numbers of rare isotope divided by the surface area. The author also gives the specific proof process of the increase of degree of rare molecule distribution. As for the gradient between surface and inner part of the droplet, the author can not give you an answer. At present, we can only assume that there is an uneven coefficient of rare isotope molecule distribution on the droplet surface.

Suppose that the radii of two droplets are $r_1$ and $r_2$ ($r_1 < r_2$), respectively. The isotope ratio in the droplet is $R$ (because the droplet is formed with the same water source, so the isotope ratio is the same), and the corresponding number of molecules is $N_1$ and $N_2$ in droplets, respectively, then the number of rare isotope molecules in the droplets are $\dfrac{R}{1+R}N_1$ and $\dfrac{R}{1+R}N_2$, respectively. Because the gradient between surface and inner part of the droplet is unknown, here we assume that there is a distribution coefficient $\alpha$ (this coefficient is independent of particle size, $0<\alpha\leqslant1$), then the degree of rare molecules distribution on surface in two droplets are $\dfrac{N_1\alpha R}{4\pi r_1^2(1+R)}$ and $\dfrac{N_2\alpha R}{4\pi r_2^2(1+R)}$, respectively.

After two droplets collide, the radius of a new droplet is denoted as $r$, and the droplet is considered to be spherical, then its radius relationship is as follows:

$$r = \sqrt[3]{r_1^3 + r_2^3} < r_1 + r_2 \quad \text{and} \quad r > r_1, r > r_2$$

The degree of rare molecules distribution on the droplet surface after collision is $\dfrac{(N_1 + N_2)\alpha R}{4\pi r^2(1+R)}$ .

According to the principle of constant droplet density, it can be shown that the following equation exists:

$$\frac{N_1}{4\pi r_1^3} = \frac{N_2}{4\pi r_2^3} = \frac{N_1 + N_2}{4\pi r^3}$$

Then there is:

$$\frac{N_1}{4\pi r_1^2} = \frac{r_1}{r_2}\frac{N_2}{4\pi r_2^2}$$

It can be proved that:

$$\frac{N_2}{4\pi r_2^2} > \frac{N_1}{4\pi r_1^2}$$

The same can be proved:

$$\frac{N_1 + N_2}{4\pi r^2} > \frac{N_2}{4\pi r_2^2}$$

That is, when the droplets aggregate, the molecular degree of rare isotopes on the surface of the droplets increases.

• Discussion

(34) Lines 365: "The calculated values" Values of what?

Response: Thanks, The calculated values refers to the calculated hydrogen and oxygen isotope values of water vapour, the sentence has been revised to "The calculated values of hydrogen and oxygen effectively captured observed variations in the water vapour isotopes"(lines 394).

(35)Lines 366-368: "The calculated values were averaged based on the number of experimental sets, and the calculated and observed water vapour isotopes during the 109–218 s period were compared (Figure 5)." Why is the model only compared to measurements after 109s? Why does the model not apply to the first period?

Response: Thanks for your comments. This calculation equation is applicable to the isotope exchange process between droplet and water vapour, as long as the isotope gradient in the exchange process has not reached the equilibrium state. This paper lists that the isotope exchange process is only concentrated in 109s-218s, because in the water vapour isotope exchange process at this stage, the droplet has undergone a complete particle size increase process, and isotope exchange process is mainly controlled by the specific surface area of the droplet in this process. At the initial time 0-109s, due to the intervention of sound waves, the degree of rare isotope molecules distribution on the surface of the droplet increases, resulting in the reversal of the isotope exchange direction, leading to droplet isotope value depletion and water vapour isotope enrichment, and effect of the variation of the degree of rare isotope molecules distribution on isotopes exchange can not calculated by IECEWV at present.

(36) What is the initial value of $\delta D$ and $\delta^{18}O$ in the model?

Response: Thanks. The initial value of $\delta D$ and $\delta^{18}O$ in the model refers to $\delta^2 H_{V0}$ and $\delta^{18}O_{V0}$ in the equation (14) and (15), which has been added on line 399.

(37)Line 375: "The mean relative errors of water vapour $\delta D$ and $\delta^{18}O$ showed" Between modeled and measured $\delta D$ and $\delta^{18}O$?

Response: Yes, it's the relative error between the model's calculated value and the observed value.

(38) Line 378: "not substantial" What do you mean by not substantial?

Response: It is intended that the expression of isotopes increment is not obvious, and we have revised the sentence to "The mean relative errors for $\delta^2 H$ and $\delta^{18}O$ showed a certain growth trend with increasing calculation time, but the increment was still not obvious"(lines 423-424).

(39) Lines 388-394: The paragraph on relative errors and their discussion is difficult to follow and the meaning of the discussion is not clear. Why is a relative error of 4.5% chosen as reference? How does the relative error help to assess the model performance?

Response: Thanks for your comments. The mean of the statistical error interval was selected as the representative value (the relative error interval is 1 %), so the cumulative frequency value of the error distribution was less than 4.5% (5%) and 9.5% (10%) is given. Of course, we also added the cumulative

frequency of the error less than 0.5% (1%). The analysis of statistical error distribution mainly shows that the model presented in this paper does not produce singular values during the whole period of the exchange process, indicating that the calculation accuracy of the model is relatively stable, we also added reasons for relative error distribution analysis which is "The relative error distribution can reflect the stability of the model in the whole exchange process"(line 434).

(40) *Line 408: "$R^2=1.0$": What is R? And if it is equal to 1, does this mean that all the data points lie perfectly on a line?*

Response:$R^2$ stands for correlation coefficient, and in order to distinguish it from the previous notation, we use the correlation coefficient instead. The mean of calculated 95 groups of water vapour isotopes is in a straight line, the author also further checked the isotope data, there is no problem. When calculating the water vapour isotope values in the exchange process, they are all affected by the reciprocal of specific surface area, in other words, the isotopic variables are all functions of the reciprocal of specific surface area, so the calculated water vapour hydrogen and oxygen isotope values are in the same straight line.

(41) *Line 411: Cluster fitting approach: What is done here? What is the meaning of the clusters? How does this prove a linear relationship between $\delta D$ and $\delta^{18}O$?*

Response: In order to explain the relationship of water vapour isotope changes after exchange, linear clusters are used. At present, meteoric water line (MWL) and evaporation line are characterized by linear relationship. The author further believes that there are linear relationships in different hydrological processes, but the current research has not given the proof process of linear relationship. The fluctuations of water vapour isotope values in the process of experimental observation lead to relatively scattered distribution of water vapour isotope values, but according to the theory, there is a certain relationship between hydrogen and oxygen isotopes, so the author adopts a linear relationship with the same slope to characterize the distribution of experimental observed values around this line. Perhaps after enough experiments and detailed experiments, we can find the linear relationship after exchange. In view of suggestion from the reviewer, the author weakens the discussion of MWL, and the modifications can be found on lines 453-458.

(42) *Line 445-447: "By integrating the Rayleigh distillation model, this approach holds significant application value for exploring the coordinated changes in precipitation and water vapour isotopes under cooling and saturated conditions." What do you mean by integrating the Rayleigh model? Is the isotope exchange not already part of the Rayleigh model through the isotopic equilibrium fractionation factor? What do you mean by cooling conditions?*

Response: As mentioned earlier, the Rayleigh fractionation model describes the evaporation or condensation process under unsaturated conditions, and isotope exchange is the process that causes equilibrium fractionation to form. When the cloud reaches saturation due to continuous cooling, the isotope value of the droplets in the cloud can be initially calculated using Rayleigh fractionation model (unsaturated state). However, if the particle size of the cloud droplets is not large enough to produce falling process after saturation, the isotope value of the cloud droplets can be calculated using this equation as the suspended droplets continue to exchange in the cloud. That is, by integrating the Rayleigh distillation model, this

approach holds significant application value for exploring the coordinated changes in precipitation and water vapour isotopes under cooling and saturated conditions

• *Conclusions*

(43)*Line 459: "the deviation of hydrogen and oxygen isotope lines from the origin" As in the abstract, what are these lines and the origin?*

Response: The line refers to meteoric water line (MWL), and the origin refers to the coordinate origin, the author has also revised the original text and can be found on line 507.

(44) *Lines 468-471: Please review the cluster approach: I don't see how the clusters prove a linear relationship between $\delta D$ and $\delta^{18}O$.*

Response: The author may not have expressed it clearly. In the original, the author used four straight lines with the same slope (different intercepts), whose intercept bandwidth was 0.027, indicating the relative position relationship between the observation point and the fitted line. The author did not fit these points, mainly to express the existence of such a phenomenon. That is, using linear clusters with the same slope may better reflect the isotopic linear relationship of water vapour after exchange.

(45) *Line 474: "the combined application of the IECEWV and Rayleigh fractionation models" Same comment as above, I don't understand what you mean by a combination of IECEWV and the Rayleigh model.*

Response: When the cloud reaches saturation due to continuous cooling, the isotope value of the droplets in the cloud can be calculated using Rayleigh fractionation model before unsaturated state. However, if the particle size of the cloud droplets is not large enough to produce falling process after saturation, the isotope value of the cloud droplets can be calculated using this equation as the suspended droplets continue to exchange in the cloud. That is, by integrating the Rayleigh distillation model, this approach holds significant application value for exploring the coordinated changes in precipitation and water vapour isotopes under cooling and saturated conditions.

• *References:*

(46) *Please, check the doi-links in the references. Many links do not work due to a punctuation mark mistake.*

Response: The author has checked all references and revised the doi-links which links do not work, but many links (the links is really true) are not available for Chinese references, so we have deleted it.

*Minor comments*

*(47) Line 78: quantitatively → do you mean "quantifying"?*

Response: The word has been revised and can be found on line 56.

*(48) Line 85: deeply → do you mean "deepen"?*

Response: The word has been revised.

*(49) Line 309: specifically → delete*

Response: The word has been revised.

*(50)Line 349: stage → do you mean "plateau"?*

Response: The word has been revised and can be found on line 382.

*(51)Line 366: in the article → I would refer to the section where you introduce the model.*

Response: The word has been revised.

*(52)Lines 444-445:"In the future, water vapour isotopic values under saturated or near-saturated conditions should be calculated without in situ observations by combining this equation with observed raindrop spectral data." → Do you mean "with in situ observations"?*

Response: Yes.

Reviewer #2

*Exchange processes between precipitation and water vapour isotopes are critical to the interpretation of isotopic signatures. In particular, with the increasing availability of isotope data from high-resolution observations, microphysical processes are becoming increasingly important in explaining isotope variations on the intra-event scale. Isotopic exchanges between water vapour and droplets under isothermal saturation conditions were investigated by theoretical calculations and experimental methods by Bai et al. This has important applications for explaining the isotopic variations of raindrops and water vapour during the actual precipitation processes.*

*Major comments*

*(1) Lines 206-207: What is the source of the water vapour that generates droplets here? Is it atmospheric water vapour in the laboratory environment?*

Response: Thanks for your comments. The droplets are formed by pressurized water flow hitting the mesh. The water source of the droplets is artificially collected river water, and the water vapour in the exchange chamber is atmospheric water vapour in experiment laboratory.

*(2) Line 251: What standard water vapour sample was utilized in the experiment? In general, liquid water standard samples are usually utilized in experiments.*

Response: Thanks. Indeed, our standard water vapour source is formed by vaporization of liquid water which is purified water. The isotope value of the standard water vapour source used for vaporization has been tested before the experiment, and this value is used as the standard value to check the concentration deviation and time drift.

*(3) Line 274: Were these liquid water samples measured for isotopic values? Variations in the isotopic values of liquid water can help explain the exchange processes.*

Response: Yes, we have tested the isotope samples of liquid water, and the isotope values are shown in the table 2, we found that the isotope values of the droplets increase after exchanged with water vapour in the exchange chamber. In addition, we conducted a significance test using one-way ANOVA, the results indicated that the droplets isotopes changed significantly after exchanged with water vapour ($p$=0.05).

Table 2 Variation of droplets isotopes after exchanged with water vapour

| Experimental number | Droplets before exchange | | | | Droplets after exchange | | | | increment | |
|---|---|---|---|---|---|---|---|---|---|---|
| | $\delta^2H$(‰) | $\delta^2H$ SD | $\delta^{18}O$(‰) | $\delta^{18}O$ SD | $\delta^2H$ (‰) | $\delta^2H$ SD | $\delta^{18}O$ (‰) | $\delta^{18}O$ SD | $\delta^2H$ (‰) | $\delta^{18}O$ (‰) |
| 1 | -43.70 | 1.53 | -7.39 | 0.33 | -42.24 | 1.65 | -7.24 | 0.17 | 1.47 | 0.14 |

| | | | | | | | | | |
|---|---|---|---|---|---|---|---|---|---|
| 2 | -43.95 | 0.95 | -7.51 | 0.23 | -41.79 | 4.29 | -7.20 | 0.61 | 2.15 | 0.31 |
| 3 | -43.14 | 0.48 | -7.51 | 0.07 | -42.12 | 1.44 | -6.36 | 0.15 | 1.02 | 1.15 |
| 4 | -44.37 | 2.61 | -7.78 | 0.38 | -40.71 | 0.80 | -6.59 | 0.14 | 3.67 | 1.18 |
| 5 | -44.47 | 1.52 | -7.63 | 0.33 | -41.45 | 2.35 | -7.24 | 0.36 | 3.01 | 0.39 |
| 6 | -43.34 | 1.39 | -7.33 | 0.27 | -42.22 | 1.16 | -6.85 | 0.15 | 1.12 | 0.48 |
| 7 | -44.19 | 1.57 | -7.78 | 0.27 | -41.16 | 1.25 | -6.98 | 0.16 | 3.02 | 0.79 |
| 8 | -43.75 | 0.59 | -7.51 | 0.11 | -40.83 | 0.57 | -7.14 | 0.11 | 2.92 | 0.38 |
| 9 | -43.57 | 0.79 | -7.41 | 0.09 | -41.98 | 0.72 | -7.18 | 0.12 | 1.59 | 0.23 |
| 10 | -43.54 | 0.15 | -7.40 | 0.10 | -40.65 | 0.89 | -7.10 | 0.14 | 2.89 | 0.29 |
| 11 | -42.76 | 2.12 | -7.40 | 0.36 | -41.71 | 1.00 | -7.36 | 0.15 | 1.04 | 0.03 |
| 12 | -42.63 | 1.02 | -7.33 | 0.13 | -41.18 | 1.60 | -7.06 | 0.20 | 1.45 | 0.27 |
| 13 | -44.34 | 1.05 | -7.63 | 0.12 | -39.95 | 0.95 | -7.02 | 0.16 | 4.38 | 0.61 |
| 14 | -43.61 | 0.71 | -7.61 | 0.07 | -40.13 | 1.03 | -7.05 | 0.22 | 3.48 | 0.56 |
| 15 | -43.52 | 1.21 | -7.56 | 0.09 | -40.36 | 1.00 | -7.01 | 0.14 | 3.16 | 0.55 |
| 16 | -42.38 | 2.57 | -7.21 | 0.42 | -39.48 | 1.03 | -6.92 | 0.13 | 2.90 | 0.29 |
| 17 | -42.30 | 1.08 | -7.44 | 0.12 | -40.39 | 1.96 | -7.10 | 0.28 | 1.91 | 0.34 |
| 18 | -41.82 | 1.75 | -7.54 | 0.29 | -42.06 | 0.69 | -7.28 | 0.13 | -0.24 | 0.26 |
| 19 | -43.75 | 0.26 | -7.33 | 0.06 | -40.30 | 0.58 | -7.16 | 0.08 | 3.45 | 0.17 |
| 20 | -43.63 | 0.65 | -7.55 | 0.17 | -40.54 | 0.26 | -7.15 | 0.03 | 3.09 | 0.40 |
| 21 | -43.72 | 1.33 | -7.99 | 0.26 | -40.49 | 0.70 | -7.09 | 0.14 | 3.23 | 0.90 |
| 22 | -43.79 | 2.38 | -7.60 | 0.33 | -43.76 | 1.34 | -7.54 | 0.23 | 0.03 | 0.07 |
| 23 | -42.56 | 0.70 | -7.11 | 0.11 | -42.05 | 1.56 | -7.28 | 0.26 | 0.51 | -0.17 |
| 24 | -41.90 | 0.33 | -7.17 | 0.04 | -42.13 | 1.15 | -7.20 | 0.13 | -0.23 | -0.03 |
| 25 | -42.91 | 0.24 | -7.21 | 0.07 | -38.89 | 1.87 | -6.57 | 0.29 | 4.02 | 0.63 |
| 26 | -44.46 | 1.05 | -7.53 | 0.14 | -38.63 | 1.59 | -6.91 | 0.27 | 5.82 | 0.62 |
| 27 | -43.62 | 3.83 | -7.45 | 0.50 | -36.88 | 1.83 | -6.79 | 0.24 | 6.74 | 0.66 |
| 28 | -43.92 | 5.44 | -7.67 | 0.79 | -42.29 | 0.95 | -7.47 | 0.14 | 1.62 | 0.21 |
| 29 | -42.35 | 3.21 | -7.31 | 0.41 | -41.40 | 5.43 | -7.16 | 0.81 | 0.95 | 0.15 |
| 30 | -43.12 | 1.01 | -7.32 | 0.13 | -40.89 | 2.59 | -6.85 | 0.35 | 2.23 | 0.47 |
| 31 | -43.36 | 4.95 | -7.27 | 0.79 | -43.36 | 2.99 | -7.89 | 0.39 | 0.00 | -0.62 |
| 32 | -44.67 | 3.72 | -7.67 | 0.40 | -42.57 | 1.81 | -7.33 | 0.27 | 2.10 | 0.34 |
| 33 | -43.89 | 3.78 | -7.38 | 0.31 | -42.52 | 1.63 | -7.01 | 0.25 | 1.37 | 0.37 |
| 34 | -43.10 | 1.10 | -7.12 | 0.13 | -41.91 | 0.70 | -7.51 | 0.13 | 1.19 | -0.39 |
| 35 | -43.05 | 2.79 | -7.43 | 0.41 | -39.37 | 0.40 | -6.86 | 0.07 | 3.68 | 0.56 |
| 36 | -42.94 | 0.96 | -7.40 | 0.13 | -40.54 | 0.85 | -7.41 | 0.15 | 2.40 | -0.02 |
| 37 | -43.35 | 1.09 | -7.41 | 0.14 | -42.89 | 0.64 | -7.78 | 0.15 | 0.46 | -0.36 |
| 38 | -40.68 | 1.92 | -6.92 | 0.24 | -43.17 | 0.86 | -7.91 | 0.14 | -2.49 | -0.99 |
| 39 | -42.05 | 2.25 | -7.09 | 0.44 | -43.42 | 1.09 | -8.04 | 0.24 | -1.37 | -0.95 |
| 40 | -40.96 | 3.70 | -7.05 | 0.35 | -41.97 | 1.01 | -7.95 | 0.16 | -1.01 | -0.90 |
| 41 | -42.77 | 2.97 | -7.35 | 0.31 | -42.02 | 1.20 | -7.73 | 0.16 | 0.76 | -0.38 |
| 42 | -43.44 | 0.92 | -7.32 | 0.09 | -41.49 | 0.82 | -7.52 | 0.19 | 1.94 | -0.19 |
| 43 | -43.23 | 0.62 | -7.32 | 0.12 | -44.73 | 1.58 | -8.34 | 0.24 | -1.50 | -1.02 |
| 44 | -38.37 | 1.75 | -6.73 | 0.35 | -43.06 | 0.34 | -7.70 | 0.06 | -4.69 | -0.97 |
| 45 | -39.11 | 3.43 | -6.85 | 0.52 | -42.10 | 1.00 | -7.39 | 0.14 | -2.99 | -0.54 |

| | | | | | | | | | |
|---|---|---|---|---|---|---|---|---|---|
| 46 | -41.69 | 1.49 | -7.11 | 0.32 | -41.55 | 1.10 | -7.41 | 0.11 | 0.14 | -0.30 |
| 47 | -38.75 | 2.82 | -6.55 | 0.27 | -44.39 | 5.88 | -8.27 | 0.92 | -5.63 | -1.72 |
| 48 | -42.52 | 5.04 | -7.35 | 0.74 | -42.22 | 0.95 | -8.20 | 0.11 | 0.30 | -0.85 |
| 49 | -43.67 | 2.86 | -7.44 | 0.36 | -43.32 | 1.09 | -7.95 | 0.14 | 0.35 | -0.51 |
| 50 | -43.40 | 2.37 | -7.45 | 0.23 | -43.00 | 1.07 | -7.87 | 0.22 | 0.41 | -0.42 |
| 51 | -43.43 | 0.31 | -7.34 | 0.07 | -42.80 | 2.99 | -7.88 | 0.46 | 0.62 | -0.54 |
| 52 | -43.29 | 0.40 | -7.36 | 0.08 | -41.52 | 1.07 | -7.72 | 0.18 | 1.77 | -0.36 |
| 53 | -42.79 | 0.64 | -7.36 | 0.13 | -42.68 | 2.98 | -7.85 | 0.41 | 0.11 | -0.48 |
| 54 | -43.58 | 1.01 | -7.48 | 0.23 | -41.76 | 1.91 | -7.80 | 0.32 | 1.82 | -0.32 |
| 55 | -43.09 | 0.33 | -7.27 | 0.05 | -43.55 | 4.40 | -8.02 | 0.64 | -0.47 | -0.75 |
| 56 | -43.05 | 0.77 | -7.25 | 0.13 | -43.72 | 2.62 | -7.84 | 0.41 | -0.66 | -0.59 |
| 57 | -43.86 | 0.33 | -7.70 | 0.03 | -44.78 | 1.22 | -7.83 | 0.15 | -0.93 | -0.13 |
| 58 | -43.01 | 1.21 | -7.38 | 0.14 | -39.08 | 0.72 | -6.71 | 0.14 | 3.93 | 0.67 |
| 59 | -43.32 | 1.10 | -7.30 | 0.12 | -38.73 | 0.41 | -6.66 | 0.08 | 4.59 | 0.63 |
| 60 | -42.24 | 0.51 | -7.54 | 0.09 | -38.53 | 0.43 | -6.73 | 0.07 | 3.70 | 0.81 |
| 61 | -42.46 | 0.72 | -7.36 | 0.03 | -38.21 | 0.95 | -6.39 | 0.17 | 4.24 | 0.97 |
| 62 | -43.06 | 1.05 | -7.43 | 0.12 | -38.64 | 0.24 | -6.70 | 0.13 | 4.41 | 0.73 |
| 63 | -43.07 | 3.56 | -7.50 | 0.58 | -38.86 | 0.64 | -6.93 | 0.17 | 4.21 | 0.57 |
| 64 | -42.88 | 0.00 | -7.46 | 0.00 | -38.71 | 0.21 | -6.47 | 0.04 | 4.18 | 0.98 |
| 65 | -41.29 | 4.33 | -7.29 | 0.58 | -38.81 | 0.99 | -6.71 | 0.12 | 2.48 | 0.59 |
| 66 | -40.57 | 0.00 | -7.33 | 0.00 | -38.14 | 0.87 | -6.75 | 0.13 | 2.43 | 0.58 |
| 67 | -42.43 | 1.10 | -7.46 | 0.11 | -37.94 | 0.87 | -6.54 | 0.13 | 4.49 | 0.92 |
| 68 | -42.79 | 2.33 | -7.44 | 0.29 | -38.47 | 0.51 | -6.62 | 0.10 | 4.32 | 0.82 |
| 69 | -42.16 | 0.64 | -7.39 | 0.12 | -38.61 | 0.53 | -6.74 | 0.15 | 3.55 | 0.65 |
| 70 | -42.32 | 4.22 | -7.46 | 0.61 | -39.04 | 0.90 | -6.86 | 0.14 | 3.28 | 0.60 |
| 71 | -41.42 | 2.99 | -7.51 | 0.41 | -38.53 | 0.83 | -6.64 | 0.16 | 2.89 | 0.87 |
| 72 | -40.88 | 4.01 | -6.98 | 0.49 | -38.54 | 0.51 | -6.66 | 0.08 | 2.34 | 0.32 |
| 73 | -42.76 | 0.59 | -7.42 | 0.10 | -39.78 | 0.23 | -6.98 | 0.02 | 2.98 | 0.44 |
| 74 | -42.51 | 0.94 | -7.49 | 0.12 | -39.32 | 0.22 | -6.79 | 0.08 | 3.19 | 0.71 |
| 75 | -43.48 | 4.98 | -7.38 | 0.68 | -39.22 | 0.60 | -6.55 | 0.09 | 4.26 | 0.83 |
| 76 | -43.52 | 2.78 | -6.96 | 0.35 | -38.32 | 0.78 | -6.89 | 0.11 | 5.20 | 0.07 |
| 77 | -43.29 | 0.97 | -6.72 | 0.21 | -37.19 | 0.55 | -6.37 | 0.11 | 6.10 | 0.36 |
| 78 | -43.81 | 0.75 | -6.91 | 0.13 | -37.74 | 0.35 | -6.42 | 0.10 | 6.07 | 0.49 |
| 79 | -44.25 | 1.83 | -6.94 | 0.20 | -39.02 | 0.33 | -6.52 | 0.12 | 5.23 | 0.42 |
| 80 | -43.99 | 2.55 | -7.06 | 0.37 | -38.23 | 0.60 | -6.55 | 0.14 | 5.76 | 0.51 |
| 81 | -44.01 | 0.83 | -7.15 | 0.12 | -38.52 | 0.47 | -6.61 | 0.09 | 5.48 | 0.54 |
| 82 | -44.28 | 0.39 | -7.39 | 0.03 | -39.32 | 0.90 | -6.74 | 0.18 | 4.97 | 0.64 |
| 83 | -44.10 | 0.02 | -7.58 | 0.06 | -38.94 | 0.08 | -6.83 | 0.04 | 5.15 | 0.75 |
| 84 | -47.23 | 0.00 | -8.30 | 0.00 | -38.88 | 0.30 | -6.57 | 0.11 | 8.35 | 1.72 |
| 85 | -45.19 | 0.00 | -7.97 | 0.00 | -39.38 | 0.35 | -6.85 | 0.10 | 5.80 | 1.12 |
| 86 | -43.68 | 0.00 | -7.82 | 0.00 | -38.82 | 1.31 | -6.79 | 0.24 | 4.86 | 1.03 |
| 87 | -40.70 | 2.09 | -7.23 | 0.23 | -39.55 | 0.55 | -6.65 | 0.09 | 1.15 | 0.58 |
| 88 | -43.02 | 3.07 | -7.72 | 0.48 | -40.24 | 0.56 | -7.30 | 0.09 | 2.79 | 0.42 |
| 89 | -43.66 | 0.00 | -7.66 | 0.00 | -39.22 | 0.57 | -7.00 | 0.07 | 4.45 | 0.66 |

| 90 | -43.14 | 1.13 | -7.63 | 0.17 | -38.54 | 2.22 | -6.73 | 0.25 | 4.60 | 0.90 |
| 91 | -44.53 | 3.62 | -7.99 | 0.61 | -40.01 | 0.73 | -6.92 | 0.15 | 4.52 | 1.07 |
| 92 | -44.10 | 1.16 | -7.71 | 0.07 | -38.96 | 0.60 | -6.70 | 0.18 | 5.15 | 1.01 |
| 93 | -45.17 | 1.41 | -7.93 | 0.18 | -38.74 | 0.70 | -6.74 | 0.12 | 6.44 | 1.19 |
| 94 | -44.66 | 1.19 | -7.61 | 0.13 | -38.59 | 0.55 | -6.36 | 0.05 | 6.08 | 1.25 |
| 95 | -44.27 | 0.49 | -7.30 | 0.14 | -38.11 | 0.51 | -6.50 | 0.11 | 6.17 | 0.80 |
| 96 | -44.36 | 0.59 | -7.03 | 0.13 | -38.35 | 0.67 | -6.60 | 0.13 | 6.02 | 0.43 |
| 97 | -41.25 | 3.25 | -6.96 | 0.55 | -38.96 | 1.24 | -6.28 | 0.27 | 2.30 | 0.67 |
| 98 | -41.45 | 1.80 | -6.88 | 0.37 | -38.62 | 0.74 | -6.43 | 0.12 | 2.83 | 0.45 |
| 99 | -43.38 | 1.33 | -6.93 | 0.10 | -38.73 | 0.43 | -6.83 | 0.07 | 4.65 | 0.10 |
| 100 | -41.51 | 1.58 | -7.10 | 0.19 | -37.97 | 1.55 | -6.37 | 0.32 | 3.55 | 0.73 |

*(4) Line 309: Figure 4 shows the water vapour isotopes within the exchange chamber, where is the water vapour coming from? Is it atmospheric water vapour inside or outside the lab? Have any comparison measurements been done to observe water vapour isotopes that have not been subjected to the exchange process?*

Response: Thanks for your comments. The initial water vapour is the original water vapour in the exchange chamber. The top of the exchange chamber is isolated from the laboratory environment by a plastic film, but the particle size test site has a hole with a diameter of 10 cm, it can be considered that the water vapour in the exchange chamber comes from the atmospheric water vapour in the experimental environment. Before the droplets was introduced into the exchange chamber, the author also observed the ambient water vapour value in the exchange chamber, as shown in the figure 3, in the figure, the gray area represents the water vapour isotope values before the exchange, it can be found that the water vapour isotopes tend to be depleted with the exchange between the droplet and the water vapour, and the mean isotopic values are -83.35 ‰- and 13.33 ‰ for $\delta^2H$ and $\delta^{18}O$ before exchange respectively, after exchange, the $\delta^2H$ and $\delta^{18}O$ are -83.54 ‰- and 13.37 ‰ respectively.

[Figure]

Figure 3 Temporal variations of water vapour isotopes before and after exchange, in the figure, the gray area represents the water vapour isotope values before the exchange.

*(5) Line 331: Are the isotopic variations shown in Figure 4 randomly selected from within the results of multiple experiments?*

Response: Thanks for your comments. The result given in Figure 4 is the average of all experimental results, that is, the average data of 95 groups of experiments. The author also added the standard errors in the figure.

*(6) Lines 369-370: The actual observations refer to the data at 0s?*

Response: Thanks. The actual observation begins when the sound wave begins to act, but the initial value of the calculation process here, that is, $\delta^2 H_{V0}$ and $\delta^{18} O_{V0}$ in the equation (14) and (15), corresponds to the data of the actual observation at 109s.

*(7) Line 400: Managave et al. (2016) indicated when the raindrops isotopically equilibrate with water vapour, the smaller drops more readily inherit higher d-excess. What is the difference between this study, which found d-excess remained constant, and previous studies?*

Response: Thanks for your valuable comments. In this study, the droplets isotope values of different sizes were not detected to compare the influence of particle size on isotope values, and the isotopic values of all droplets that gathered together were given in the paper, so the conclusions are not contradictory. In addition, the particle size of the droplet studied in this paper is relatively small compared with the size of the external field, and its average particle size is about 30 μm, which belongs to a particle size range. Therefore, the *d*-excess value remains unchanged, which is consistent with the existing research. If it is assumed that both large droplets and small droplets are formed by water vapour condensation (not considering the case of large droplets formed by the collision of small droplets), the condensation formation time of small droplets is longer than that of large droplets. Then, according to Rayleigh fractionation model, the *d*-excess value of small droplets is higher than that of large droplets. The results of this paper are used to explain the phenomenon that droplet size is negatively correlated with *d*-excess under equilibrium conditions. According to the above proof, when droplet size increases, the degree of rare isotope molecules distribution on the surface of large droplet increases, and to maintain equilibrium with the surrounding water vapour isotope gradient, the isotope value of large droplet will be depleted compared with that of small droplet. This process can be regarded as the inverse process of evaporation, and the *d*-excess in the small droplet is higher than that in the large droplet.

We also found that *d*-excess of droplets decrease slightly after sedimentation, and *d*-excess decreased by 0.4 (The average values of *d*-excess before and after exchange were 15.88‰ and 15.48‰, respectively) after exchange, which can basically be considered as the same *d*-excess (Bai et al., 2021). Recently, the author also consulted relevant literature, only one isotope value was given in the exchange of liquid droplets-water vapour and snow-water vapour, so it is can not calculate the change of *d*-excess. However, both theoretical and experimental observations show that the *d*-excess of the exchange process is constant, so this conclusion is credible.

Bai, W., Wei, J., Shi, Y., Zhao, Z., and Li, Q.: Microphysical Characteristics and Environmental Isotope Effects of the Micro-Droplet Groups under the Action of Acoustic Waves, Atmos., 12, 1488, https://doi,org/10.3390/atmos12111488, 2021b.

Friedman, I., Machta, L., and Soller, R.: Water-vapor exchange between a water droplet and its environment, J. Geophys. Res., 67, 2761-2766, 1962.

Wahl, S., Steen-Larsen, H. C., Reuder, J., & Hörhold, M.: Quantifying the stable water isotopologue exchange between snow surface and lower atmosphere by direct flux measurements. Journal of Geophysical Research: Atmospheres,126, 2021, e2020JD034400. https://doi.org/10.1029/2020JD034400